# Ant venoms contain vertebrate-selective pain-causing sodium channel toxins

Samuel D. Robinson [1] ✉, Jennifer R. Deuis[1], Axel Touchard [2,3], Angelo Keramidas[1], Alexander Mueller [1], Christina I. Schroeder [1,11], Valentine Barassé [3], Andrew A. Walker [1], Nina Brinkwirth[4], Sina Jami[1], Elsa Bonnafé [3], Michel Treilhou [3], Eivind A. B. Undheim[1,5,6], Justin O. Schmidt[7,8], Glenn F. King [1,9] & Irina Vetter [1,10] ✉

Stings of certain ant species (Hymenoptera: Formicidae) can cause intense, long-lasting nociception. Here we show that the major contributors to these symptoms are venom peptides that modulate the activity of voltage-gated sodium (Na$_V$) channels, reducing their voltage threshold for activation and inhibiting channel inactivation. These peptide toxins are likely vertebrate-selective, consistent with a primarily defensive function. They emerged early in the Formicidae lineage and may have been a pivotal factor in the expansion of ants.

The majority of ant species produce venom[1] and many are capable of delivering a painful sting. Nonetheless, the chemistry and pharmacology underpinning ant stings has remained largely unexplored, not least due to their relatively small size, which has made venom collection and analysis technically challenging. Another reason is the widespread misconception that all ants have simple acid-containing venoms. While it is true that ants of the subfamily Formicinae spray formic acid[2] (or other acids), it is now known that the venoms of most other ant species are composed primarily of peptides[3].

Most ant and other aculeate hymenopteran venom peptides appear to be derived from a large gene superfamily known as the aculeatoxins[4]. Of the aculeatoxins studied so far, most have an amphipathic structure and a capacity to disrupt biological membranes[4–6]. Membrane disruption by these amphipathic peptide toxins may be responsible for the pain of some hymenopteran stings[7], including certain ants[4,5]. However, these observations do not satisfactorily explain the long-lasting and characteristic sting symptoms of some ant species. For example, stings of both *Tetramorium africanum* (subfamily Myrmicinae) of

tropical Western Africa and the greenhead ant, *Rhytidoponera metallica* (subfamily Ectatomminae) of Australia can produce severe and long-lasting local pain accompanied by piloerection, hyperhidrosis and an axon reflex flare at the sting site (personal observations, S.D.R. and A.T.). Stings of the bullet ant, *Paraponera clavata* (subfamily Paraponerinae), of tropical Central and South America, can cause uncontrollable trembling and immediate intense local pain that lasts for hours[8,9], and is famously exploited by certain tribes of the Amazonas in their initiation rituals[10].

While the mechanisms underlying the characteristic sting symptoms of *T. africanum* and *R. metallica* remain to be investigated, the dramatic symptoms of *P. clavata* stings motivated a series of studies on this ant's venom in the early 1990s. Piek and colleagues identified a 25-residue disulphide-free peptide, poneratoxin[11] (Table. 1), which modulated sodium currents in frog and rat skeletal muscle fibres[12] via what appeared to be a previously undescribed mode of action[13]. More recently, it was reported that poneratoxin modulates the mammalian voltage-gated sodium (Na$_V$) channel subtype Na$_V$1.7[14]. Because Na$_V$1.7 serves a critical role in mammalian pain signalling[15], its

[1]Institute for Molecular Bioscience, University of Queensland, Saint Lucia QLD 4072, Australia. [2]CNRS, UMR Ecologie des forêts de Guyane (EcoFoG), Campus Agronomique; BP 316, 97379 Kourou, Cedex, France. [3]Equipe BTSB-EA 7417, Université de Toulouse, Institut National Universitaire Jean-François Champollion; Place de Verdun, 81012 Albi, France. [4]Nanion Technologies, Munich 80339, Germany. [5]Centre for Ecological and Evolutionary Synthesis, Department of Biosciences, The University of Oslo, Oslo, Norway. [6]Centre for Advanced Imaging, University of Queensland, Saint Lucia QLD 4072, Australia. [7]Southwestern Biological Institute, Tucson AZ 85745, USA. [8]Department of Entomology, University of Arizona, Tucson AZ 85721, USA. [9]Australian Research Council Centre of Excellence for Innovations in Peptide and Protein Science, University of Queensland, Saint Lucia QLD 4072, Australia. [10]School of Pharmacy, University of Queensland, Woolloongabba QLD 4102, Australia. [11]Present address: Genentech, 1 DNA Way, South San Francisco 94080 CA, USA.
✉e-mail: sam.robinson@uq.edu.au; i.vetter@imb.uq.edu.au

modulation by poneratoxin was proposed as the mechanism underlying the painful stings of *P. clavata*[14].

In this study, we tested the hypothesis that the characteristic painful stings of *T. africanum*, *R. metallica* and *P. clavata* are due to peptide toxins, including poneratoxin. We demonstrate that the venoms of these and other ant species contain peptide toxins with activity at mammalian Na$_V$ channels, an effect that likely mediates the nociceptive effects of these venoms. These toxins evolved early in the Formicidae, most likely as vertebrate-selective defensive agents.

## Results

### Peptide toxins with activity at mammalian voltage-gated sodium channels are primarily responsible for the intense long-lasting pain of T. africanum stings

We hypothesised that ant stings cause nociception in mammals primarily via direct local action on the peripheral sensory nervous system. To identify the pain-causing agent(s) underlying stings of *T. africanum*,

we fractionated venom using reversed-phase high-performance liquid chromatography (RP-HPLC) and, using calcium imaging, tested the capacity of each fraction to activate mammalian sensory neurons of the mouse neuroblastoma × rat dorsal root ganglion (DRG) F11 cell line. Three fractions (f12, f22 and f23) activated F11 cells (Fig. 1a, b): application of f12 caused a rapid increase in intracellular calcium ([Ca$^{2+}$]$_i$) which then decreased back below baseline−consistent with membrane disruption by e.g. pore formation, while application of f22 and f23 caused a rapid and sustained increase in [Ca$^{2+}$]$_i$. We determined the primary structure of two of these fractions by searching liquid chromatography tandem mass spectrometry (LC-MS/MS) data against a *T. africanum* venom-gland transcriptome. Fraction 12 contained a 21-residue cysteine-free peptide that we called Ta2a, while f23 contained a 29-residue cysteine-free peptide that we called Ta3a (Table 1). Ta2a is similar (59% sequence identity) to the previously described M-MYRTX-Tb1a (also known as bicarinalin), a membrane-active peptide from venom of the ant *T. bicarinatum*[16] (Supplementary Fig. 1), whereas Ta3a (precursor and mature peptide) is similar to several uncharacterised peptide sequences from other ant species (presented herein). We were unable to determine the primary structure of the active component in f22, but its similar activity and RP-HPLC elution time suggest it is related to Ta3a.

We chemically synthesised Ta2a and Ta3a and determined their potency for activation of F11 cells. Ta3a caused calcium influx in F11 cells with an EC$_{50}$ of 5.8 ± 1.6 nM, while Ta2a was almost 3000-fold less potent with an EC$_{50}$ of 19.7 ± 2.0 μM (Fig. 1c). Next, we tested the capacity of each peptide to cause nocifensive behaviours in mice.

**Table 1 | Pain-causing Na$_V$ channel toxins of ant venoms**

|       | Species                  | Primary structure              |
|-------|--------------------------|--------------------------------|
| Pc1a  | *Paraponera clavata*     | FLPLLILGSLLMTPPVIQAIHDAQR*     |
| Ta3a  | *Tetramorium africanum*  | LAPIFALLLLSGLFSLPALQHYIEKNYIN* |
| Rm4a  | *Rhytidoponera metallica*| FPPLLLLAGLFSLPALQHYIETKWIN*    |
| Mri1a | *Manica rubida*          | GLPLLALLMTLPFIQHAITN*          |

*C-terminal amidation.

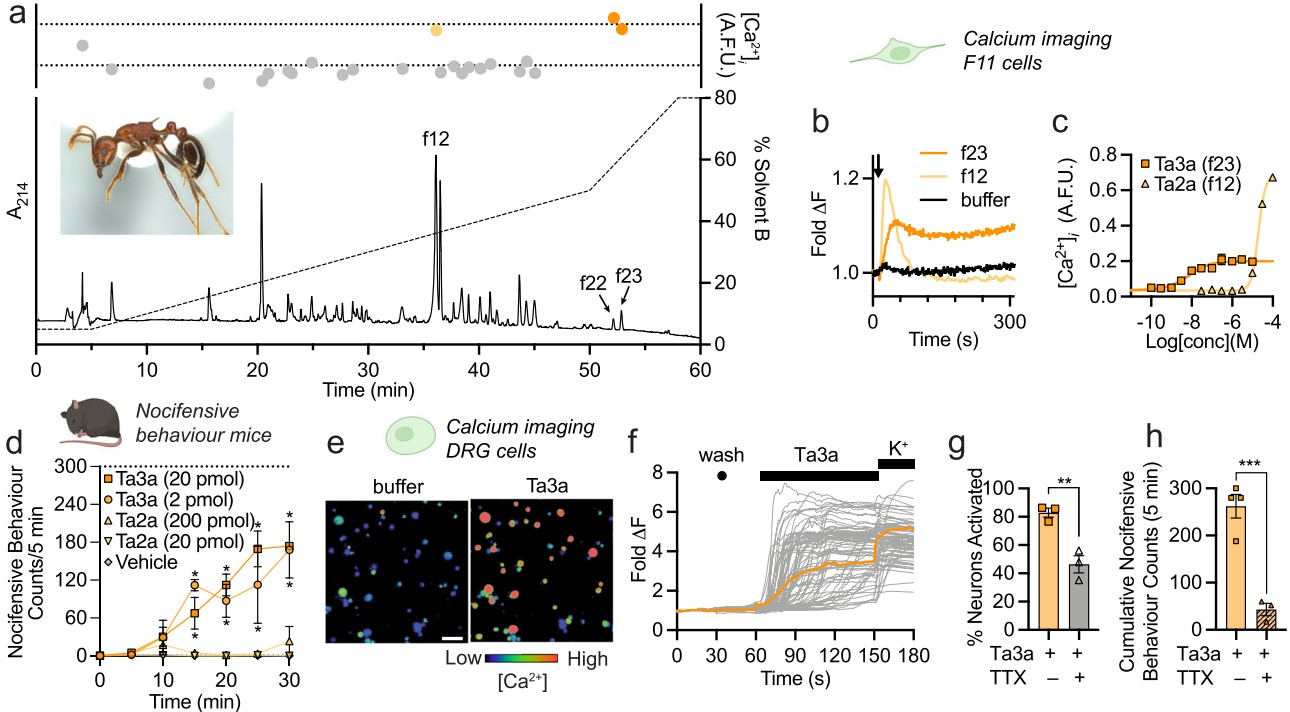

**Fig. 1 | A peptide toxin, Ta3a, causes the intense long-lasting pain of *Tetramorium africanum* stings. a** Chromatogram showing fractionation of *T. africanum* venom using RP-HPLC and activity of fractions on F11 cells (data are mean of two duplicates). Active fractions are coloured orange and peaks are labelled. Inset: *T. africanum* worker (~ 5 mm in length). **b** [Ca$^{2+}$]$_i$ in F11 cells after addition of f12 and f23 (data are mean of two duplicates). The arrow indicates the time of addition. **c** Potency of synthetic Ta3a and Ta2a in F11 cells, as monitored by changes in [Ca$^{2+}$]$_i$ ($n$ = 3 independent experiments). **d** Spontaneous nocifensive behaviours in mice following shallow intraplantar injection of Ta3a (2 or 20 pmol; $n$ = 3 mice per group). Ta2a was inactive up to a dose of 200 pmol. *$P$ < 0.05 (two-way ANOVA with Holm-Šídák's multiple-comparisons to negative control). **e** Representative (of 3

independent experiments) pseudocolour images illustrating [Ca$^{2+}$]$_i$ in DRG neurons before (buffer) and after application of Ta3a (100 nM); scale bar = 100 μm. **f** Time course of individual DRG neuron responses to Ta3a. Each trace represents an individual neuron and the orange trace represents the average response; K$^+$, 30 mM KCl (positive control) (representative of 3 independent experiments). **g** Percentage of DRG neurons activated by Ta3a (100 nM) in the absence or presence of TTX (1 μM). **$P$ = 0.0059 (unpaired $t$-test; two-sided; $n$ = 3 independent experiments). **h** Cumulative spontaneous nocifensive behaviours in mice after intraplantar injection of saline ($n$ = 4 mice) or TTX (2 μM) ($n$ = 3 mice), 30 min after injection of Ta3a (60 pmol). ***$P$ = 0.0010 (unpaired $t$-test; two-sided). Data are expressed as mean ± SEM. Source data are provided as a Source Data file.

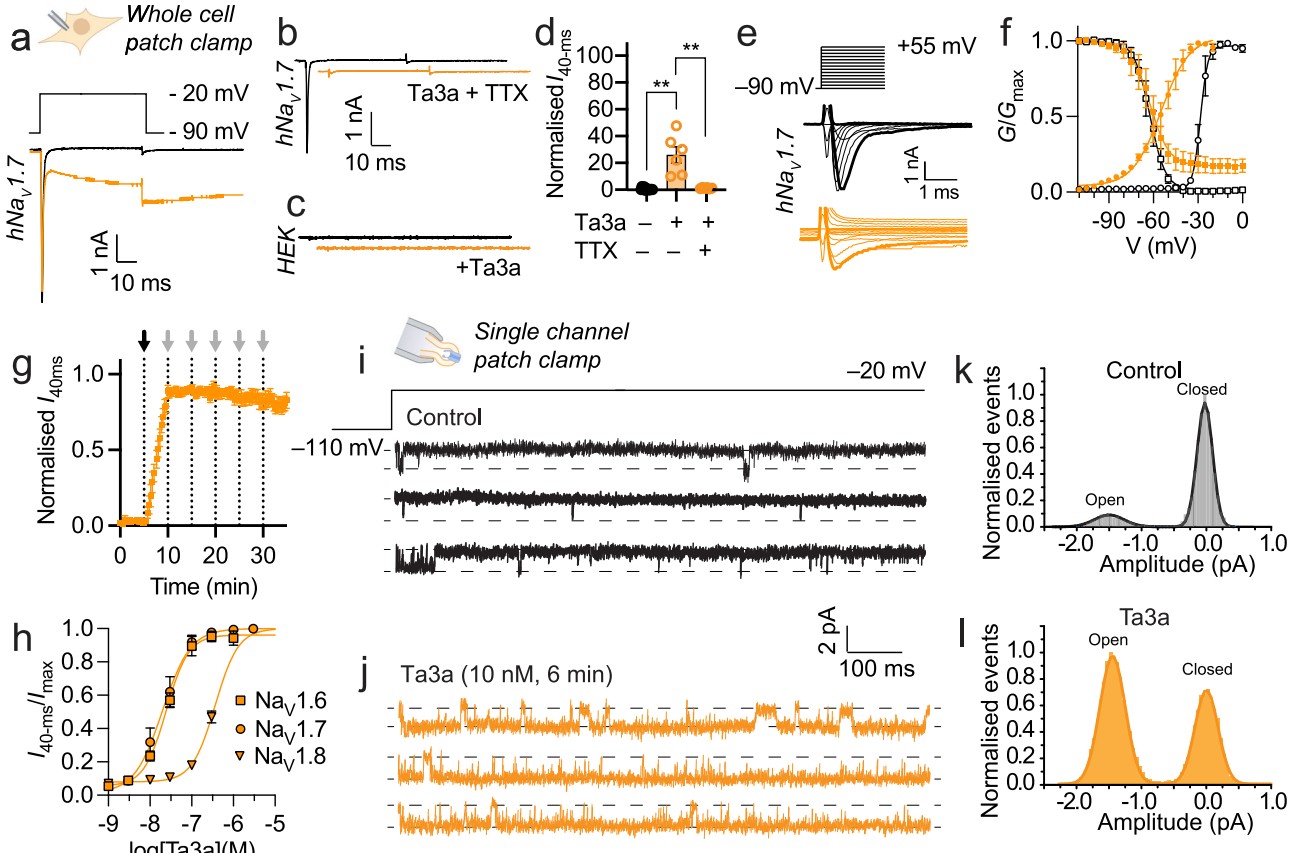

**Fig. 2 | Ta3a is a potent modulator of mammalian Na_V channels.**
**a** Representative hNa_V1.7 current response to a step depolarisation from −90 to −20 mV in the absence (black) and presence of Ta3a (30 nM, orange).
**b** Representative hNa_V1.7 current response to a step depolarisation from −90 to −20 mV in the absence (black) and presence of Ta3a (30 nM) + TTX (1 μM, orange).
**c** Representative HEK293 cell current response to a step depolarisation from −90 to −20 mV in the absence (black) and presence of Ta3a (30 nM, orange).
**d** hNa_V1.7 sustained current (current amplitude at 40-ms) evoked by a step depolarisation from −90 to −20 mV before (control) and after treatment with Ta3a (30 nM) or Ta3a + TTX (1 μM), expressed as % of control $I_{peak}$. **P = 0.0073 (Ta3a *versus* vehicle), P = 0.0020 (Ta3a *versus* Ta3a + TTX) (unpaired *t*-test, two-sided; n = 6 cells). **e** Representative traces from hNa_V1.7 *I-V* experiments before (top) and after addition of Ta3a (10 nM, bottom). Traces corresponding to −20 mV step are bold. **f** hNa_V1.7 *G-V* (circles) and SSFI (squares), before (white) and after addition of

Ta3a (10 nM, orange). Data for Ta3a are sustained current ($I_{40-ms}$) (n = 3 cells). **g** Washout protocol measuring Ta3a-induced hNa_V1.7 persistent current over time (n = 6 cells). The addition of Ta3a (3 nM) and washouts (ECS) are indicated by black and grey arrows, respectively. **h** Concentration-response relationships for Ta3a modulation of human Na_V1.6 (n = 8 cells), Na_V1.7 (n = 7 cells) and Na_V1.8 (n = 4 cells), where response is [(sustained current ($I_{40-ms}$)/peak current ($I_{max}$)]. **i** Three representative sweeps of single hNa_V1.7 channel currents in response to a voltage step to −20 mV (from a holding potential of −110 mV). Channel openings are downward deflections. **j** Representative sweeps of single-channel currents after perfusing patch with Ta3a (10 nM, 6 min). **k, l** Normalised all-point amplitude histograms of activity showing relative level of open-state events after Ta3a exposure (n = 5 patches) compared to control (n = 6 patches). Data are expressed as mean ± SEM. Source data are provided as a Source Data file.

Consistent with *T. africanum* stings in humans (personal observations, A.T.), intraplantar injection of Ta3a (2 or 20 pmol) caused gradually-developing and long-lasting nocifensive behaviours (Fig. 1d). Ta2a was inactive up to a dose of 200 pmol. Thus, we concluded that Ta3a (and likely a second closely-related peptide (f22)) is primarily responsible for the long-lasting nociception of *T. africanum* stings.

To investigate the cellular and molecular mechanisms by which Ta3a causes nociception in mammals, we tested its activity on primary cultures of dissociated mouse DRG cells, which include sensory neurons responsible for the detection of painful stimuli. Application of Ta3a (100 nM) to mouse DRG cells caused an immediate and sustained increase in fluorescence indicative of increased $[Ca^{2+}]_i$ in 82.9 ± 3.2% of neurons (Fig. 1e–g). This was reduced to 46.4 ± 6.1% of neurons (P = 0.0059, unpaired *t*-test) in the presence of 1 μM tetrodotoxin (TTX), a blocker of mammalian Na_V channels, (Fig. 1f), indicating that Ta3a can directly activate mammalian sensory neurons and that its mode of action in this setting involves TTX-sensitive Na_V channel subtypes. Consistent with TTX-sensitive Na_V channel subtypes playing a key role in the pain-causing effect of Ta3a, Ta3a-induced

spontaneous nocifensive behaviours in mice were largely ameliorated by injection of 2 μM TTX (P = 0.0010, unpaired *t*-test; Fig. 1h).

Using whole-cell voltage-clamp electrophysiology, we investigated the effects of Ta3a on mouse Na_V1.7. A representative mNa_V1.7 current response to a step depolarisation of the cell membrane voltage from −90 to −20 mV in the absence and presence of Ta3a (30 nM) is shown in Supplementary Fig. 2. Ta3a, with a half-maximal effective concentration ($EC_{50}$) of 18 ± 5 nM, converted the normally rapidly-activating and inactivating Na_V1.7 current into one which does not inactivate even after repolarisation to −90 mV (Supplementary Fig. 2a–c; Supplementary Table 1). The effects of Ta3a (30 nM) on human Na_V1.7 were comparable (sustained current of 26.0 ± 6.0% of control peak; P = 0.0073, paired *t*-test; $EC_{50}$ of 30 ± 9 nM; Fig. 2a) and remaining experiments were performed at human channels. The "leak" current elicited by Ta3a is via the central pore of the Na_V channel (as opposed to another channel or via direct pore-formation by the peptide) as evidenced by the absence of inward current in hNa_V1.7-expressing cells in the presence of TTX (1 μM) (P = 0.0020, unpaired *t*-test; Fig. 2b, d) and the absence of inward current upon addition of

Ta3a to untransfected HEK293 cells (Fig. 2c). Furthermore, $Cs^+$, present in the intracellular recording solution (for all electrophysiology experiments in this study, whole cell and single channel), precluded the involvement of potassium channels endogenously expressed in HEK293 cells. Ta3a (10 nM) induced a strong hyperpolarising shift in the voltage-dependence of $hNa_V1.7$ activation ($V_{50}$ shifted from $-28.3 \pm 0.3$ mV to $-54.8 \pm 1.1$ mV, $P < 0.0001$, unpaired $t$-test), while voltage-dependence of steady-state fast inactivation (SSFI) was unaffected (Fig. 2e, f). Ta3a (3 nM)-induced $hNa_V1.7$ persistent current was very slowly reversible with repeated wash steps over 30 min (Fig. 2g).

We extended our analysis to include human $Na_V$ channel subtypes $Na_V1.6$ (TTX-sensitive), $Na_V1.8$ and $Na_V1.9$ (TTX-resistant), which are also expressed in mammalian DRG neurons and have each been implicated in peripheral pain signalling[15,17]. $hNa_V1.6$ was as sensitive to Ta3a as $hNa_V1.7$, with an $EC_{50}$ of $25 \pm 2$ nM, whereas $Na_V1.8$ was an order of magnitude less sensitive ($EC_{50}$ of $331 \pm 58$ nM) (Fig. 2h; Supplementary Table 1). Ta3a was also active at $Na_V1.9$, where at a concentration of 1 μM it inhibited inactivation ($I_{50\text{-ms}} = 106.9 \pm 10.2$ % of peak versus $41.5 \pm 3.1$ % of control peak; $P < 0.0001$, paired $t$-test; Supplementary Fig. 3).

Next, we investigated the effects of Ta3a on single $hNa_V1.7$ channel current amplitude and conductance. To elicit single channel activation, membrane voltage was stepped from a holding potential of −110 mV, where $Na_V$ channels are closed, to a voltage of −20 mV. Under control conditions, this produced current records comprising of brief, dispersed single-channel active periods, separated by relatively longer quiescent periods corresponding to channel inactivation (Fig. 2i, k). After recording channel activity under control conditions, the recorded patches were then continuously perfused with 10 nM Ta3a before applying the same activating voltage step. These recordings revealed a marked enhancement of single-channel activity in the presence of the toxin (Fig. 2j, l). While Ta3a did not alter the single-channel amplitude (−1.46 ± 0.02 pA (Ta3a-treated) versus −1.49 ± 0.02 pA (control); $P = 0.350$, unpaired $t$-test), it increased the proportion of open channel events from $13.1 \pm 0.3$% (control) to $60.1 \pm 0.2$% (Ta3a-treated) ($P = 0.008$, unpaired $t$-test).

Taken together, these data demonstrate that the peptide toxin Ta3a contributes to the long-lasting nociception from stings of *T. africanum*, an effect that can be explained by its modulation of $Na_V$ channels in peripheral sensory neurons, although we cannot at present exclude contribution(s) of other pharmacological activities.

## Pain-causing toxins that modulate mammalian sodium channels are present in the venoms of other ant species

The pharmacological activity and primary structure of Ta3a shared sequence features with poneratoxin from the venom of *P. clavata* (Table 1), leading us to hypothesise that: (i) poneratoxin, as previously suggested[9,14], is responsible for the intense long-lasting pain of *P. clavata* stings and (ii) pain-causing $Na_V$ channel toxins may be present in other ant venoms.

We synthesised poneratoxin (Pc1a) and tested its ability to induce nocifensive behaviours in mice. Shallow intraplantar injection of Pc1a (20 to 600 pmol) in mice caused immediate, long-lasting and near-maximal (i.e. 300 counts/5 min) nocifensive behaviours (Fig. 3a), consistent with symptoms following *P. clavata* stings in humans. Given that the venom of *P. clavata* is composed near-exclusively of poneratoxin (Pc1a or paralogues thereof) (Supplementary Fig. 4)[11,18], these data suggest that this peptide is likely the primary agent underpinning the severe and long-lasting pain from stings of *P. clavata*.

Application of Pc1a (500 nM) to mouse DRG cells caused an immediate and sustained increase of $[Ca^{2+}]_i$ in $84.9 \pm 7.6$% of neurons, which in the presence of TTX was reduced to $3.5 \pm 1.2$% of neurons ($P = 0.0004$, unpaired $t$-test; Supplementary Fig. 5). Pc1a-induced spontaneous nocifensive behaviours in mice were largely ameliorated by co-injection of 2 μM TTX ($P < 0.0001$, unpaired $t$-test;

Supplementary Fig. 5). These data are consistent with our hypothesis that nociception from *P. clavata* stings is due to the direct action of poneratoxin on TTX-sensitive $Na_V$ channels in peripheral sensory neurons.

Consistent with the report by Johnson et al., Pc1a was active at $hNa_V1.7$ ($EC_{50} = 2.3 \pm 0.4$ μM), where at a concentration of 3 μM it caused a sustained current (12.2 ± 1.2% of control peak), a reduction in peak current amplitude (to $57.8 \pm 4.9$% of control) and a hyperpolarising shift in $V_{50}$ from $-25.1 \pm 1.5$ mV to $-56.4 \pm 12.6$ mV (Fig. 3b, c; Supplementary Fig. 6a, b). Both the *G-V* and SSFI relationships of Pc1a-treated $Na_V1.7$ fitted to a double-Boltzmann distribution. Pc1a had comparable effects on mouse $Na_V1.7$ (Supplementary Fig. 2; Supplementary Table 1). Pc1a (3 μM)-induced $hNa_V1.7$ persistent current was slowly reversible with repeated wash steps over 30 min (Fig. 3d).

We found that $hNa_V1.6$ was more sensitive than $hNa_V1.7$ to Pc1a, both in terms of toxin potency ($EC_{50} = 97 \pm 10$ nM) and the magnitude of the effects on current (Fig. 3e–h, Supplementary Fig. 5c, d, Supplementary Table 1). Conversely, $hNa_V1.8$ was less sensitive ($EC_{50} > 10$ μM). At a concentration of 1 μM, Pc1a inhibited $hNa_V1.9$ inactivation ($I_{50\text{-ms}} = 58.6 \pm 4.4$ % of peak versus $45.0 \pm 2.2$ % of peak for control, $P = 0.0032$, paired $t$-test; Supplementary Fig. 3).

We identified several peptides in other ant venoms with similar primary structure to Ta3a and Pc1a (Table 1). Of these, we synthesised Rm4a from the venom of the Australian greenhead ant *R. metallica* and $U_3$-MYTX-Mri1a (Mri1a) from the venom of the European fire ant *Manica rubida*. Intraplantar injection of Rm4a (20 to 200 pmol) in mice caused dose-dependent spontaneous nocifensive behaviours, which, consistent with sting symptoms in humans (personal observation, S.D.R.), were gradual in onset, reaching near maximal at 30 min post-injection (200 pmol) (Fig. 3i). Application of Rm4a (500 nM) to DRG cells caused an immediate and sustained increase in $[Ca^{2+}]_i$ in $82.5 \pm 10.4$% of neurons, which in the presence of 1 μM TTX was reduced to $48.1 \pm 0.8$% of neurons ($P = 0.00297$, unpaired $t$-test; Supplementary Fig. 7). Rm4a-induced spontaneous nocifensive behaviours in mice were reduced by injection of 2 μM TTX ($P = 0.0175$, unpaired $t$-test; Supplementary Fig. 5). Together, these data demonstrate that Rm4a likely contributes to the long-lasting pain from stings of *R. metallica* and that its mode-of-action involves modulation of TTX-sensitive $Na_V$ channels (and possibly one or more other TTX-resistant ion channels or receptors) in peripheral sensory neurons.

Under whole-cell voltage-clamp in HEK293 cells expressing $hNa_V$ channels, Rm4a exhibited similar activity to Pc1a, converting the normally rapidly activating and inactivating $hNa_V1.7$ current into one that does not inactivate (sustained current of $33.1 \pm 2.4$% of control; $P < 0.0001$, paired $t$-test; Fig. 3j; Supplementary Fig. 7a, b), and caused a hyperpolarising shift in the voltage-dependence of channel activation ($\Delta V_{50} = -24.4 \pm 9.5$ mV, $P = 0.0227$, unpaired $t$-test; Fig. 3j, k), with comparable effects on mouse $Na_V1.7$ (Supplementary Fig. 2, Supplementary Table 1). Similar to Pc1a, both the *G-V* and SSFI relationships of Rm4a-treated $hNa_V1.7$ fitted to a double-Boltzmann distribution. Rm4a (3 μM)-induced $hNa_V1.7$ persistent current was slowly reversible with repeated wash steps over 30 min (Fig. 3l).

$EC_{50}$ values for modulation of $hNa_V1.6$, $hNa_V1.7$ and $hNa_V1.8$ by Rm4a were $196 \pm 23$ nM, $1.9 \pm 0.4$ μM and $8.4 \pm 1.0$ μM, respectively (Fig. 3m, Supplementary Table 1). The effects of Rm4a on $hNa_V1.6$ currents were similar to those for $hNa_V1.7$, although greater in amplitude (at the same toxin concentration) (Fig. 3m–p; Supplementary Fig. 7c, d). At a concentration of 1 μM, Rm4a inhibited $hNa_V1.9$ inactivation ($I_{50\text{-ms}} = 53.3 \pm 4.2$% of peak versus $40.5 \pm 2.0$% of peak for control, $P < 0.0087$, paired $t$-test; Supplementary Fig. 3).

Mri1a (20 pmol to 2 nmol) also induced dose-dependent spontaneous nocifensive behaviours in mice, although it was less potent than the other peptides (Supplementary Fig. 8a), and caused an immediate

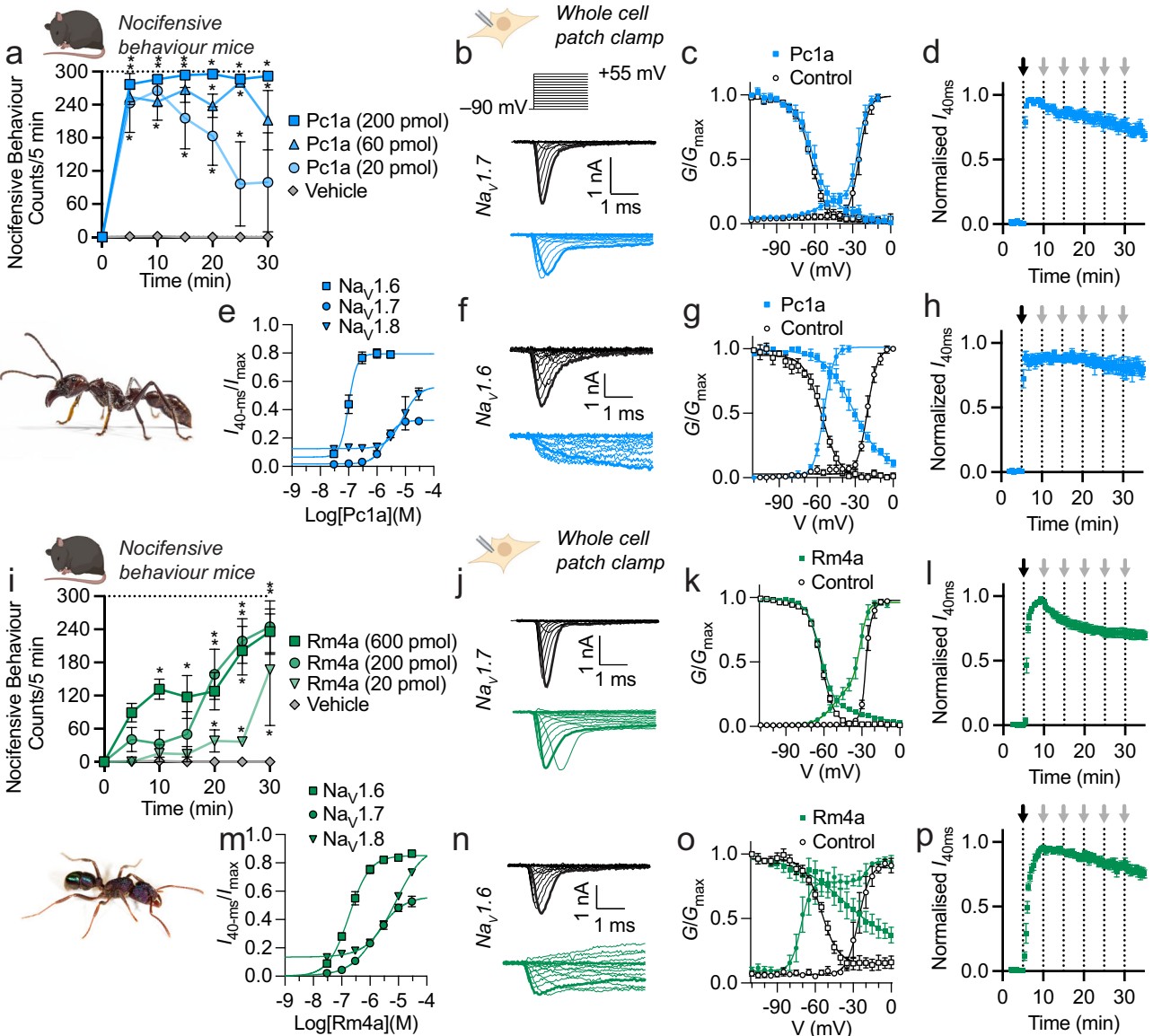

**Fig. 3 | Pain-causing toxins that modulate Na_V channels are present in the venoms of other ant species. a** Spontaneous nocifensive behaviours in mice following shallow intraplantar injection of Pc1a ($n = 3$ mice per group). *$P < 0.05$ (two-way ANOVA with Holm-Šídák's multiple-comparisons to negative control). Inset: *P. clavata* worker (~20 mm in length). Photo credit: Hadrien Lalagüe.
**b** Representative *I-V* traces for hNa_V1.7 channels expressed in HEK293 cells before (top) and after addition of Pc1a (3 μM, bottom). Traces corresponding to −20 mV steps are bold. **c** hNa_V1.7 *G-V* (circles) and SSFI (squares) curves, before (white) and after addition of Pc1a (3 μM, blue) ($n = 7$ cells). **d** Washout protocol measuring Pc1a-induced hNa_V1.7 persistent current over time ($n = 4$ cells). The addition of Pc1a (3 μM) and washouts (ECS) are indicated by black and grey arrows, respectively.
**e** Concentration-response relationship for Pc1a modulation of human Na_V1.6

($n = 4$), Na_V1.7 ($n = 8$) and Na_V1.8 ($n = 3$), where response was [sustained current ($I_{40-ms}$)/peak current ($I_{max}$)]. **f** Representative *I-V* traces for hNa_V1.6 before (top) and after addition of Pc1a (3 μM, bottom). Traces corresponding to −20 mV steps are bold. **g** hNa_V1.6 *G-V* (circles) and SSFI (squares) curves, before (white) and after addition of 3 μM Pc1a (blue) ($n = 5$ cells). **h** Washout protocol measuring Pc1a-induced hNa_V1.6 persistent current over time ($n = 4$ cells). The addition of Pc1a (3 μM) and washouts (ECS) are indicated by black and grey arrows, respectively. **i–p** Equivalent data for Rm4a. Inset: *R. metallica* worker (~7 mm in length). **i** ($n = 3$ mice per group). *$P < 0.05$ (two-way ANOVA with Holm-Šídák's multiple-comparisons to negative control). **k** ($n = 7$ cells), **l** ($n = 4$ cells), **m** ($n = 4–7$ cells), **o** ($n = 6$ cells), **p** ($n = 5$ cells). Data are expressed as mean ± SEM. Source data are provided as a Source Data file.

and sustained increase of $[Ca^{2+}]_i$ in $52.1 \pm 4.6\%$ of neurons in mouse DRG cell cultures, which in the presence of TTX was reduced to $2.9 \pm 2.4\%$ of neurons ($P = 0.0007$, unpaired *t*-test) (Supplementary Fig. 8). Under whole-cell voltage-clamp in HEK293 cells expressing hNa_V channels, Mri1a exhibited similar activity to Ta3a. At hNa_V1.7, Mri1a (3 μM) caused a small sustained current ($3.4 \pm 0.7\%$ of control peak; $P = 0.0061$, paired *t*-test) which remained after repolarisation, a reduction in peak current amplitude (to $68.1 \pm 10.6\%$ of control, $P = 0.0394$, paired *t*-test), and a hyperpolarising shift in the voltage-dependence of activation ($\Delta V_{50} = -8.5 \pm 0.4$, $P < 0.0001$, unpaired *t*-test; Supplementary Fig. 8). hNa_V1.6 was more sensitive to Mri1a

($EC_{50} = 3.3 \pm 0.4$ μM) than hNa_V1.7 ($EC_{50} > 10$ μM) and, at the concentrations tested, Mri1a was inactive at hNa_V1.8 and hNa_V1.9 (Supplementary Figs. 3a and 8; Supplementary Table 1).

## Ta3a, Rm4a, Pc1a and Mri1a are vertebrate-selective defensive toxins

Many ant species use their venoms for both defence and predation (i.e., to incapacitate arthropods that they feed to their larvae). While the capacity of Ta3a, Pc1a, Rm4a and Mri1a to cause nocifensive behaviours in mammals suggested a defensive role for these peptides, we were interested in whether they had specifically evolved as

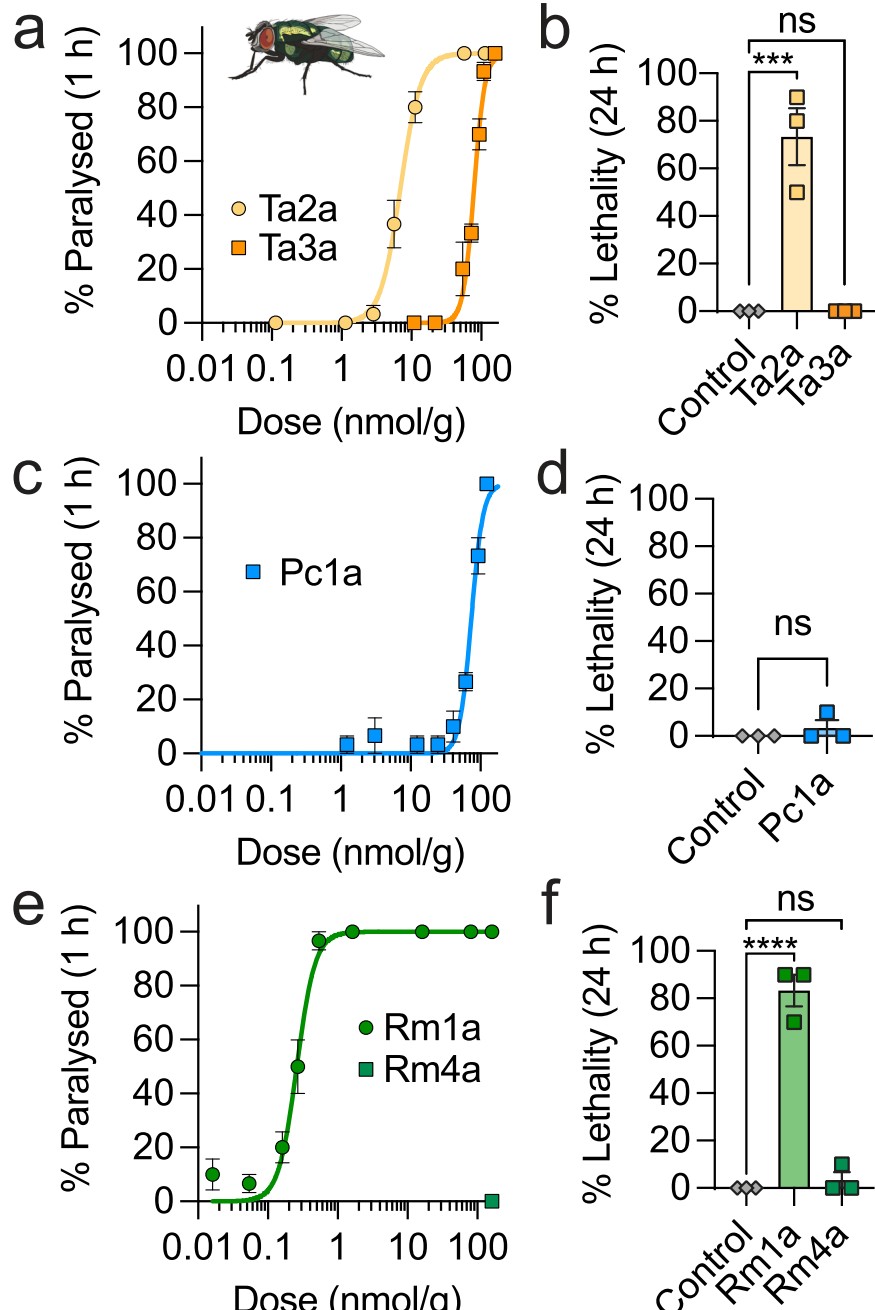

**Fig. 4 | Ta3a, Pc1a, Mri1a and Rm4a are vertebrate-selective defensive toxins.**
**a** Injection of Ta2a and Ta3a in blowflies caused paralysis with $PD_{50}$ [1 h] of 7.1 and 77.8 nmol/g, respectively (percentage of flies paralysed at 1 h post-injection; $n = 3$ independent experiments). **b** Ta2a (157 nmol/g) was lethal to blowflies (****$P = 0.0006$; one-way ANOVA with Dunnett's multiple comparison test; $n = 3$ independent experiments), while Ta3a (109 nmol/g) was not (percentage lethality recorded 24 h after injection). **c** Pc1a caused paralysis with $PD_{50}$ [1 h] of 74.3 nmol/g ($n = 3$ independent experiments), **d** and (up to 123 nmol/g) was not lethal ($n = 3$

independent experiments). **e** Rm1a caused paralysis with $PD_{50}$ [1 h] of 0.27 nmol/g, while Rm4a was inactive (up to 200 nmol/g) ($n = 3$ independent experiments). **f** Rm1a (162 nmol/g) was lethal (****$P < 0.0001$; one-way ANOVA with Dunnett's multiple comparison test; $n = 3$ independent experiments), while Rm4a (100 nmol/g) was not ($n = 3$ independent experiments). Data are expressed as mean ± SEM and (in panels **a**, **c**, **e**) fitted with a nonlinear regression with variable slope (four parameters). Source data are provided as a Source Data file.

defensive agents under selection pressure from vertebrate predators, or whether their activity in vertebrates was secondary to a predatory function. To examine this question, we tested the insecticidal activity of each peptide, alongside other peptides from each venom, by intrathoracic injection into blowflies (*Lucilia caesar*).

Intrathoracic injection of Ta3a and Pc1a in blowflies caused a slowly-developing temporary paralysis at high doses ($PD_{50}$[1 h] = 77.8 ± 1.4 (Ta3a) and 74.3 ± 2.1 nmol/g (Pc1a); Fig. 4a, c), and neither peptide was lethal at the highest dose tested (109 nmol/g

(Ta3a), 123 nmol/g (Pc1a)) (Fig. 4b, d). Intrathoracic injection of Rm4a (up to a dose 200 nmol/g) had no effect on blow flies (Fig. 4e, f), consistent with our previous study showing that intrathoracic injection of Rm4a in fruit flies (*Drosophila melanogaster*) and house crickets (*Acheta domesticus*) had no effect at doses up to 40 nmol/g[19]. Similarly, Mri1a has previously been shown to be inactive on intrathoracic injection in blowflies[20].

For comparison, intrathoracic injection of Ta2a (from *T. africanum* venom) in blowflies caused immediate contractile paralysis

$((PD_{50})[1\ h] = 7.1 \pm 0.9$ nmol/g), which was lethal at doses > 57 nmol/g (Fig. 4a, b). Another peptide from the venom of *R. metallica* (Rm1a)[19], caused immediate, irreversible paralysis and lethality in blowflies with a $PD_{50}[1\ h]$ of $0.27 \pm 0.03$ nmol/g (Fig. 4e, f), consistent with our previous study showing immediate paralysis and lethality in both fruit flies ($LD_{50} = 0.17$ nmol/g; $PD_{50}[5\ min] = 0.023$ nmol/g) and crickets ($PD_{50}[30\ min] = 0.3$ nmol/g). Several peptides from the venom of *M. rubida* cause paralysis and/or death in blowflies[20].

To place these potencies in perspective, the amount of Ta3a required to cause delayed, weak and temporary paralysis in blowflies was more than 2000-fold higher (~4500 pmol ($PD_{50}$ equivalent)) than the amount required to cause long-lasting spontaneous nocifensive behaviours in mice (2 pmol). By contrast, Ta2a from the same venom caused immediate, irreversible paralysis in blowflies with a $PD_{50}$ equivalent to an amount of ~380 pmol, but had no effect in mice at 200 pmol. For Pc1a, the amount required to cause weak, slowly-developing and temporary paralysis in blowflies was more than 220-fold higher (4580 pmol ($PD_{50}$ equivalent)) than the amount which caused immediate near-maximal spontaneous nocifensive behaviours in mice (20 pmol). Rm4a, which was inactive in insects (tested up to 3800 pmol), caused long-lasting spontaneous nocifensive behaviours in mice at 20 pmol, while Rm1a, a different peptide from the same venom, caused immediate, irreversible paralysis in blowflies with a $PD_{50}$ equivalent to ~5 pmol, but had no effect in mice at 200 pmol.

Taken together, these data suggest that different peptides in the venoms of ants have evolved under different selection pressures to have specialised functions − i.e., to rapidly incapacitate and/or kill invertebrate prey or to defend against vertebrate predators − and that the function of the $Na_V$ channel toxins Ta3a, Pc1a, Rm4a and Mri1a in their respective venoms is primarily, if not exclusively, the latter.

### Evolution of $Na_V$ channel toxins in Formicidae

To investigate the evolutionary relationship between $Na_V$ channel toxins found in the venom of different Formicidae species (and potentially other Hymenoptera), we used the complete precursor sequences of Ta3a, Pc1a, Rm4a and Mri1a as queries to search the National Centre for Biotechnology Information (NCBI) nonredundant (NR) and Transcriptome Shotgun Assembly (TSA) databases and Hymenoptera Genome Database[21]. No similar sequences were detected outside of the Formicidae. Within the Formicidae, related sequences were detected in species of the subfamilies Myrmicinae, Ectatomminae and Myrmeciinae, each in the formicoid lineage, but, other than poneratoxin from *P. clavata*, no related sequences were detected in poneroid ants. Several of the detected sequences have been reported in venom-gland transcriptomes and some of the corresponding mature peptides have been found in the venom of the respective species. A sequence alignment is shown in Fig. 5a. The mature peptides of $U_3$-MYRTX-Tb1a-c from the venom of *T. bicarinatum* are similar to those characterised in this study and likely share similar $Na_V$ channel modulatory activity, although this remains to be experimentally confirmed. By contrast, while the related sequences in species of the subfamily Myrmeciinae were relatively highly expressed as transcripts in the venom glands of these ants, the coding regions of both sequences were interrupted by one or more stop codons (Mc5a and Mg10a; Fig. 5a). We also found no evidence of their predicted mature peptides in the respective venoms, leading us to conclude that these genes have been pseudogenised in Myrmeciinae. Thus, this class of $Na_V$ channel toxins are present in the venoms of representatives of subfamilies known to cause long-lasting and characteristic sting symptoms e.g. Paraponerinae, Myrmicinae and Ectatomminae, and are absent from the venoms of representatives of subfamilies that cause "short and sharp" sting symptoms[9] e.g. Myrmeciinae and Ponerinae. A most-parsimonious character state reconstruction using accelerated transformation optimisation suggested that $Na_V$ channel toxins have evolved on two independent occasions in the Formicidae−once early

in the formicoid lineage and again independently in Paraponerinae (poneratoxin) (Fig. 5b).

The precursor sequence and architecture of each of the $Na_V$ channel toxins characterised in this study are consistent with that of other peptides of the hymenopteran aculeatoxin gene superfamily[4]. A phylogenetic reconstruction of the aculeatoxins grouped all ant venom $Na_V$ channel toxins into a single, well-supported monophyletic clade, with the exception of poneratoxin (Supplementary Fig. 9). Another hymenopteran $Na_V$ channel toxin, α-pompilidotoxin, reported in the venom of the pompilid wasp *Anoplius samariensis*[22,23], grouped separately to the ant venom $Na_V$ channel toxins. These data are suggestive of two separate origins of $Na_V$ channel toxins in the venoms of the Formicidae and another separate origin in pompilid wasps. On each occasion, the $Na_V$ channel toxins appear to be derived from membrane-targeting aculeatoxin peptides.

### Discussion

Ants are one of the most successful animal groups, constituting 15−20% of the terrestrial animal biomass[24]. The evolution of eusociality is considered instrumental to their success, allowing them to exploit and defend a wider range of resources[25]. But eusociality also has costs, not least the potential nutritional bounty that the colony's brood (eggs, larvae and pupae) represents to vertebrate predators. Thus, a prerequisite for the success of ants is their ability to defend their colonies from much larger vertebrate predators, which is achieved, in many species, by the capacity to deliver a painful defensive sting[26].

Certain ant species have characteristic and exceptionally painful stings[9]. Here we demonstrated that this symptomatology can be explained by the presence of peptide toxins that modulate vertebrate $Na_V$ channels. These toxins appear to have evolved on two occasions in the Formicidae−once, early in the formicoid lineage prior to their major diversification, and sometime later in the poneroid lineage, possibly exclusively in Paraponerinae. One can envisage how ants could have benefited from such a weapon by improving their defensive capabilities against tetrapod predators, and how the evolution of a toxin that could cause intense pain in vertebrates could have contributed to the successful diversification of the ants in the Cretaceous.

$Na_V$ channels are transmembrane proteins that regulate the influx of $Na^+$ ions. In excitable cells they play an essential role in action potential initiation and propagation, and are integral to neuronal and muscle function in animals. Consequently, numerous and diverse toxins have evolved to interfere with $Na_V$ channel function[27−29]. For example, α-scorpion toxins, δ-conotoxins and "sea anemone toxins", are peptides from the venoms of scorpions, cone snails and sea anemones, respectively, that bind to "sites 3 or 6" of the voltage-sensing domain(s) of $Na_V$ channels and inhibit or delay channel inactivation. β-scorpion toxins, another class of scorpion venom peptides, bind to "site 4" of the voltage-sensing domain(s) of $Na_V$ channels and promote channel opening. Several small hydrophobic alkaloids, including batrachotoxin (found in poison frogs of the genus *Phyllobates* and poison birds of the genera *Pitohui* and *Ifrita*), veratradine (from the root of Veratrum plants) and aconitine (from the plant *Acotinum napellus*), bind to "site 2" and have more complex effects on $Na_V$ channels. The ant venom $Na_V$ channel toxins are structurally and mechanistically distinct from other peptidic $Na_V$ channel modulators, with their complex effects on $Na_V$ channels more closely resembling those caused by the small hydrophobic alkaloids. While activity at mammalian $Na_V$ channels, specifically inhibition of inactivation, is sufficient to explain the painful nature of stings by *T. africanum*, *R. metallica*, and *P. clavata*, it should be noted that we cannot, at present, exclude pharmacological activity at other targets that may additionally contribute to enhanced neuronal excitability.

The repeated evolution of $Na_V$ channel toxins in hymenopteran venoms may be testament to the importance of $Na_V$ channels in vertebrate nociception, and the Formicidae can now be included in the

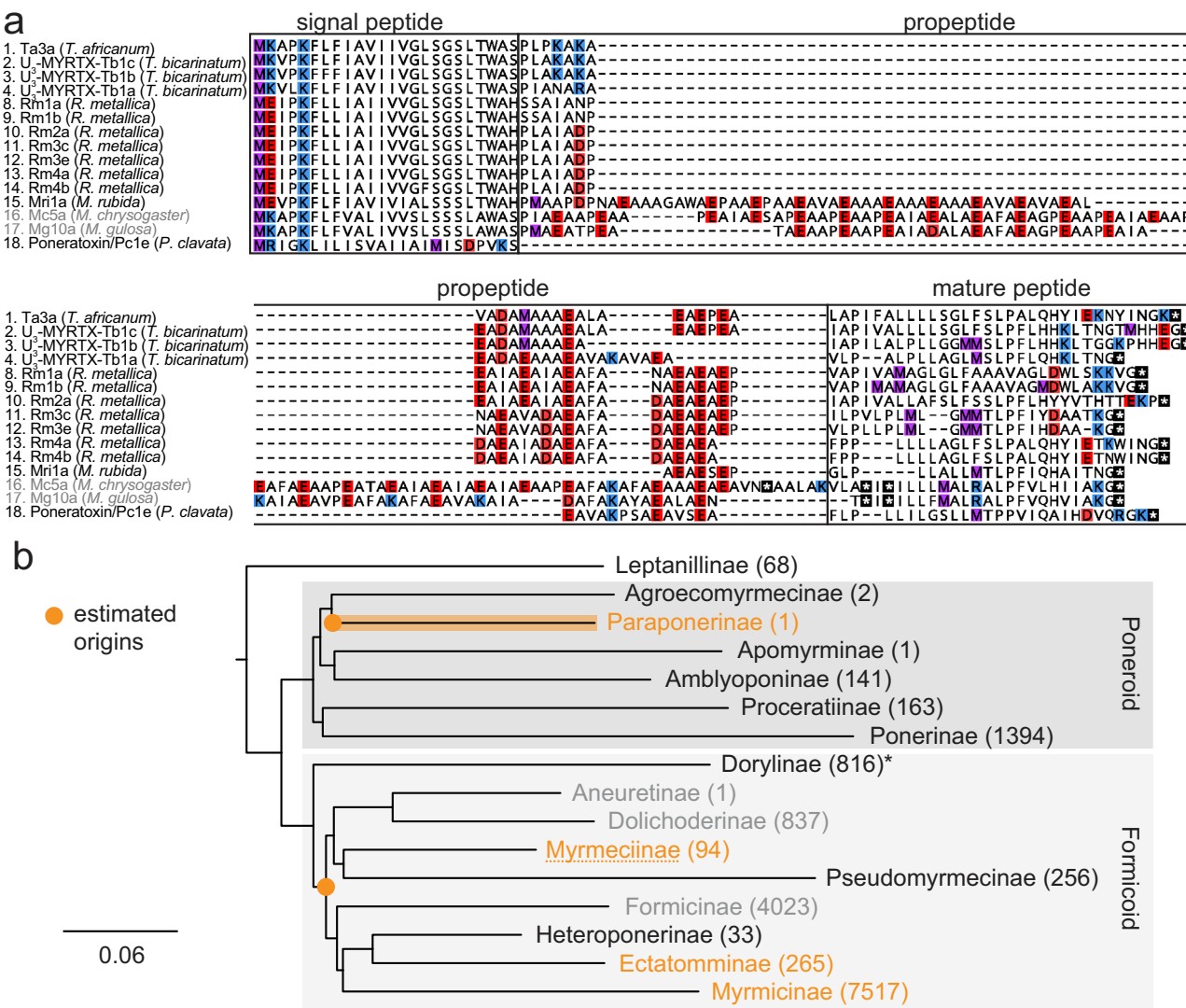

**Fig. 5 | Evolution of Na$_V$ channel toxins in ant venoms. a** Alignment of the precursor sequences of Ta3a, Poneratoxin (Pc1e), Rm4a and Mri1a with peptides detected in ant venoms (or ant venom-gland transcriptomes). Mc5a and Mg10a (labelled grey) are pseudogenes. Methionine, lysine/arginine, aspartate/glutamate, and cysteine residues are highlighted in purple, blue, red, and yellow, respectively. Stop codons are indicated by asterisks. Post-translational modifications are not shown. **b** Simplified phylogeny of the Formicidae (based on ref. 34), illustrating the diverse assemblage of organisms in which Na$_V$ channel toxins have evolved, in this case, as defensive agents against vertebrates. estimated origins of ant-venom Na$_V$ channel toxins. Subfamilies in which ant-venom Na$_V$ channel toxin sequences were detected are highlighted in orange. Non-stinging subfamilies are coloured grey. Estimated origins of ant-venom Na$_V$ channel toxins are shown in orange. The number of described extant species are given in parentheses alongside each subfamily name. Note that subfamily Dorylinae, highlighted with an asterisk, includes both stinging and non-stinging species.

diverse assemblage of organisms in which Na$_V$ channel toxins have evolved, in this case, as defensive agents against vertebrates.

## Methods

### Assay–guided fractionation of *T. africanum* venom

*T. africanum* venom (250 µg) was fractionated via RP-HPLC using a Phenomenex Gemini NX-C18 column (250 × 4.6 mm; particle size, 3 µm; pore size, 110 Å) with a gradient of 5–50% solvent B (90% acetonitrile (ACN), 0.05% TFA) over 45 min at a flow rate of 1 mL/min. 23 fractions were collected based on absorbance at 214 nm. Fractions were dried by vacuum concentration and resuspended in 50 µL of pure water, from which 1-µL aliquots were used (in a final volume of 30 µL) for calcium imaging experiments.

F11 (mouse neuroblastoma × DRG neuron hybrid) were maintained on Ham's F12 media supplemented with 10% FBS, 100 µM hypoxanthine, 0.4 µM aminopterin, and 16 µM thymidine (Hybri-Max, Sigma Aldrich). 384-well imaging plates (Corning, Lowell, MA, USA) were seeded 24 h prior to calcium imaging, resulting in ~90%

confluence at the time of imaging. Cells were loaded for 30 min at 37 °C with Calcium 4 assay component A in physiological salt solution (PSS; 140 mM NaCl, 11.5 mM D-glucose, 5.9 mM KCl, 1.4 mM MgCl$_2$, 1.2 mM NaH$_2$PO$_4$, 5 mM NaHCO$_3$, 1.8 mM CaCl$_2$, 10 mM HEPES) according to the manufacturer's instructions (Molecular Devices, Sunnyvale, CA). Ca$^{2+}$ responses were measured using a FLIPR$^{TETRA}$ fluorescent plate reader equipped with a CCD camera (Ex: 470 to 490 nm, Em: 515 to 575 nM) (Molecular Devices, Sunnyvale, CA). Signals were read every second for 10 s before, and 300 s after, the addition of venom fractions (in PSS supplemented with 0.1% BSA).

### Peptide synthesis

Pc1a, Rm4a and Mri1a were produced using Fmoc solid-phase peptide synthesis at 0.1 mmole scale. Protecting groups used were Lys/Trp/His(Boc), Ser/Thr/Tyr(tBu), Asp/Glu(OtBu), Asn/Gln/Cys/His(Trt), and Arg(Pbf). Peptides were assembled on Rink-amide ProTide resin (CEM, Matthews, NC) to produce the native C-terminal amide. Peptides were assembled on a CEM Liberty Prime HT24 microwave synthesiser (CEM

Corporation, Matthews, NC, USA) using *N,N*'-diisopropylcarbodiimide (DIC)/oxyma. Fmoc groups were removed with 20% pyrrolidine, as per manufacturer protocols.

Peptides were released from resin by treatment with 95% trifluoroacetic acid (TFA)/2.5% water/2.5% triisopropylsilane (TIPS) for 30 min at 40 °C on a CEM Razor (CEM Corporation). Peptides were precipitated with 15 mL ice-cold ether, extracted in 50/50 v/v mixture of Solvent A (0.05% TFA) and Solvent B (90% ACN, 0.045% TFA) and lyophilised prior to purification. Peptides were purified via RP-HPLC using an Agilent Zorbax 300SB-C18 column (150 × 21.2 mm; particle size, 5 μm) on a LC-20AP HPLC system (Shimadzu Corp.) with a gradient of 20–70% B over 50 min at a flow rate of 16 mL/min. Fraction purity was assessed using electrospray ionisation mass spectrometry (ESI-MS), and analytical RP-HPLC. Pure fractions were lyophilised, pooled, and stored at room temperature until use.

Ta2a and Ta3a were purchased from GenScript Biotech corporation (Netherlands) with a purity of 99.1% and 95.4%, respectively.

Stock solutions of Ta3a, Pc1a, Rm4a and Mri1a were prepared by dissolving lyophilised peptide first in 100% DMSO then diluting to 1 mM peptide, 5% dimethyl sulfoxide (DMSO) (v/v) in $H_2O$ (the peptides were not soluble in 100% $H_2O$ at a concentration of 1 mM). The stock solution of Ta2a was prepared by dissolving the lyophilised peptide in 100% $H_2O$.

## Nocifensive behaviour experiments

Male 5–8 week old C57BL/6 J mice used for behavioural experiments were purchased from the Animal Resources Centre (WA, Australia). They were housed in groups of up to four per cage, maintained on a 12/12 h light-dark cycle (19–21 °C, 60–70% humidity), and fed standard rodent chow and water *ad libitum*. Peptides diluted in saline containing 0.1% bovine serum albumin (BSA; Sigma-Aldrich) were administered in a volume of 20 μL into the hindpaw by shallow intraplantar injection. Negative-control animals were injected with saline containing 0.1% BSA. Following injection, spontaneous nocifensive behaviour events were counted from 30 min video recordings by a blinded experimenter. For analysis of spontaneous nocifensive behaviours, a two-way ANOVA with Holm-Šídák's multiple-comparisons test was used to test difference to negative control over the time course of the experiment and an unpaired *t*-test was used to test for differences in the sum of nocifensive behaviour counts at 30 min between treated and negative-control animals.

In the experiments testing for amelioration of nocifensive behaviours by TTX, Pc1a (60 pmol) was administered as above, with or without TTX (2 μM), and spontaneous nocifensive behaviour events were counted from 5 min video recordings by a blinded experimenter. For Ta3a and Rm4a, which had a slower onset of action, peptides (60 pmol) were administered as above, and at 30 min, mice were injected in the same paw with either saline or TTX (2 μM) and spontaneous nocifensive behaviour events were counted from 5 min video recordings by a blinded experimenter.

Experiments involving animals were approved by The University of Queensland Animal Ethics Committee (UQ AEC approval numbers PHARM/526/18 and 2021/AE000448).

## Calcium imaging assay of mammalian sensory neurons

DRG cells were isolated from 5–8-week-old male C57BL/6 mice purchased from the Animal Resources Centre. DRGs were dissociated, then cells plated in Dulbecco's Modified Eagle's Medium (DMEM; Gibco, MD, USA) containing 10% foetal bovine serum (FBS) (Assaymatrix, VIC, Australia) and penicillin/streptomycin (Gibco) on a 96-well poly-D-lysine-coated culture plate (Corning, ME, USA) and maintained overnight. Cells were loaded with Fluo-4 AM calcium indicator, according to the manufacturer's instructions (ThermoFisher Scientific, MA, USA). After loading (1 h), the dye-containing solution was replaced with assay solution (Hanks' balanced salt solution, 20 mM HEPES).

Images were acquired at X10 objective at 1 frame/s (excitation 485 nm, emission 521 nm). Fluorescence corresponding to $[Ca^{2+}]_i$ of ~200 cells per experiment was monitored in parallel using an Nikon Ti-E deconvolution inverted microscope, equipped with a Lumencor Spectra LED Lightsource. Baseline fluorescence was monitored for 30 s. At 30 s, assay solution was replaced with either assay solution, or assay solution containing TTX (1 μM), then at 1 min with test peptide (in assay solution ± TTX) and monitored for 1.5 min before being replaced with assay solution and then KCl (30 mM; positive control). Experiments involving use of mouse tissue were approved by the UQ AEC (approval TRI/IMB/093/17).

## Whole-cell voltage-clamp electrophysiology

HEK293 cells stably expressing the α-subunit of mouse Nav1.7, human $Na_V1.6$ or Nav1.7 plus the β1 subunit (SB Drug Discovery, Glasgow, United Kingdom) and Chinese Hamster Ovary (CHO) cells stably expressing human $Na_V1.8$ plus the β3 subunit in a tetracycline-inducible system (ChanTest, Cleveland, OH, United States), were cultured were maintained on MEM supplemented with 10% heat-inactivated FBS, 2 mM L-glutamine in an incubator at 37 °C with 5% $CO_2$ and passaged every 3–4 days (at 70–80% confluency) using TrypLE Express (Thermo Fisher Scientific).

Whole-cell patch-clamp experiments were performed using a QPatch16X automated electrophysiology platform (Sophion Bioscience, Ballerup, Denmark). The extracellular solution (ECS) contained 145 mM NaCl (replaced with 70 mM choline chloride for $Na_V1.7$), 4 mM KCl, 2 mM $CaCl_2$, 1 mM $MgCl_2$, 10 mM HEPES, and 10 mM glucose (pH 7.4; osmolarity, 305 mOsm). The intracellular solution (ICS) contained 140 mM CsF, 1 mM/5 mM EGTA/CsOH, 10 mM HEPES, and 10 mM NaCl (pH 7.3) with CsOH (osmolarity, 320 mOsm). Peptides were diluted in ECS with 0.1% BSA.

Concentration-response experiments were performed using a holding potential of −90 mV and a 50-ms pulse to −20 mV (+10 mV for $Na_V1.8$) every 20 s (0.05 Hz). *I-V* curves were obtained with a holding potential of −90 mV followed by a series of 500-ms step pulses that ranged from −110 to +55 mV in 5-mV increments (repetition interval, 5 s) before and after 5-min incubation with peptide. Conductance-voltage curves were obtained by calculating the conductance (*G*) at each voltage (*V*) using the equation $G = I/(V - V_{rev})$, where $V_{rev}$ is the reversal potential, and they were fitted with the following single or double Boltzmann equations: $I = I_{max}/\{1 + \exp[(V_{50} - V_m)/k]\}$ or $I = \{I_{max(a)}/[1+\exp(\{V_{50\,(a)} - V_m\}/k_{(a)})]\} + \{I_{max(b)}/[1 + \exp(\{V_{50(b)} - V_m\}/k_{(b)})]\}$, where $I_{max}$ is the maximal current after normalisation to the driving force, $V_{50}$ is the half-activation potential, $V_m$ is the membrane potential, and *k* is the slope factor. For statistical comparison of *G-V* curves, a two-tailed unpaired *t*-test was used.

Single-channel recordings were made at room temperature using the excised outside-out patch configuration. Currents were recorded with an EPC 10 USB Heka Patch Clamp Amplifier (HEKA, Elekronik), filtered (−3 dB, 4-pole Bessel) at 5 kHz, and sampled at 50 kHz using PatchMaster software. Currents were elicited by applying 20–50 voltage steps to −20 mV from an initial holding voltage of −110 mV. The step frequency was 0.5 Hz. Patch electrodes were made from borosilicate glass capillaries (G150F-3; Warner Instruments, CT, USA) and heat-polished to a final resistance of 5–12 MΩ when filled with intracellular solution. The intracellular solution contained 145 mM CsCl, 2 mM $MgCl_2 6H_2O$, 2 mM $CaCl_2 2H_2O$, 10 mM HEPES and 5 mM EGTA, adjusted to pH 7.4 with CsOH. The extracellular solution contained 140 mM NaCl, 5 mM KCl, 2 mM $CaCl_2 2H_2O$, 1 mM $MgCl_2 6H_2O$, 10 mM HEPES and 10 mM D-glucose, adjusted to pH 7.4 with NaOH.

For human $Na_V1.9$ (Icagen, Durham, NC, USA), HEK293 cells stably expressing the α-subunit of $hNa_V1.9$ were maintained on MEM supplemented with 10% heat-inactivated FBS. Whole-cell patch-clamp experiments were performed using a SynchroPatch 384 automated electrophysiology platform (Nanion Technologies, Munich, Germany).

The ECS contained 140 mM NaCl, 4 mM KCl, 2 mM CaCl$_2$, 1 mM MgCl$_2$, 5 mM Glucose, 10 mM HEPES and 100 nM TTX (pH 7.4). The ICS contained 110 mM CsF, 10 mM CsCl, 10 mM NaCl, 10 mM HEPES and 10 mM EGTA (pH 7.2). Peptides were diluted in ECS with 0.1% BSA. Experiments were performed using a holding potential of −130 mV and a 100-ms pulse to −40 mV every 20 s (0.05 Hz) before and after a 4.5-min incubation with 1 μM peptide.

## Insecticidal assay

Blowflies (*Lucilia caesar*; 1-4 d post-emergence; average mass 19 mg) were injected into the ventrolateral thorax with 1 μL of negative-control solution (water or 5% DMSO) or peptide (in water or 5% DMSO) using a 1 mL Hamilton syringe (1000 Series Gastight™, Hamilton company, Reno, USA) with a fixed 29-gauge needle. Flies were assessed for paralysis and/or lethality immediately, and at 1 h and 24 h post-injection. For each toxicity assay, up to seven doses of each peptide (*n* = 10 flies per dose) and the appropriate negative control (*n* = 10 flies) were used. Each assay was repeated three times.

## Phylogenetic analysis

For the phylogenetic reconstruction of the aculeatoxins (Supplementary Fig. 9), complete precursor sequences were aligned using the L-INS-i algorithm of MAFFT v7.309[30]. We selected the most appropriate evolutionary model (JTTDCMut+G4) using ModelFinder[31] before using IQ-TREE v2.0.6[32] to reconstruct the molecular phylogeny by maximum likelihood. Branch support values were estimated by ultrafast bootstrap using 10,000 replicates[33].

For the phylogeny of the Formicidae (Fig. 5b), we used the "ant-tree-101taxa-raxml-partitioned-byhcluster" dataset of Branstetter et al.[34], and collapsed branches to subfamily level. We also generated a trimmed phylogeny containing only the seven sampled sub-families (Paraponerinae, Ponerinae, Dorylinae, Myrmeciinae, Formicinae, Ectatomminae, Myrmicinae), which we used to estimate the origins of ant venom Na$_V$ channel toxins in the Formicidae (Fig. 5b) along with a data matrix consisting of one character with two states (presence or absence of venom Na$_V$ channel toxin) for each sub-family. PAUP*[35] v4.0a, build 168, was used to reconstruct Na$_V$ channel aculeatoxin recruitments and losses using the accelerated transformation (ACCTRAN) parsimony character optimisation algorithm (Supplementary Data 1). ACCTRAN was chosen because it infers character transformations as close as possible to the base of the tree and thereby represents the most conservative approach in terms of the estimated number of times Na$_V$ channel toxins have evolved.

## Statistics

Data were plotted and analysed using Prism v9.0.0 (GraphPad Software, San Diego, CA, USA). Statistical significance was defined as *P* < 0.05. All data are presented as mean ± SEM.

## Reporting summary

Further information on research design is available in the Nature Portfolio Reporting Summary linked to this article.

## Data availability

The data that support this study are available from the corresponding authors upon request. Sequences of Ta3a, Ta2a, Pc1a, Rm4a and Mri1a are available in Genbank: OW518818.1, Genbank: OW518839.1, UniProt: P41736, GenBank: MW317032 and GenBank: MN765042.1, respectively. The source data underlying Figs. 1–4 and Supplementary Figs. 2, 3, 5–8 are provided as a Source Data file. Source data are provided with this paper.

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

## Acknowledgements

The Australian Research Council (Discovery Project DP210102425 to S.D.R, J.R.D., and J.O.S., Centre of Excellence grant CE200100012 to G.F.K., and Future Fellowship FT160100055 to C.I.S). National Geographic Society early career grant (EC-58468R-19 to S.D.R.). J.R.D., G.F.K. and I.V. are supported by Australian National Health & Medical Research Council (NHMRC) fellowships (APP1139961, APP1136889 and APP1162503, respectively). A.T was supported by an Investissement d'Avenir grant (ANR-10-LABX-25-01) from Agence Nationale de la Recherche and a Fonds européen de développement régional grant (GY0013708). E.A.B.U. is supported by the Norwegian Research Council (FRIPRO-YRT Fellowship no. 287462) and the European Research Council (ERC Starting Grant 101039862). We thank Nanion Technologies, in particular Nadine Becker, Alison Obergrussberger and Ilka Rinke-Weiss, for making the experiments on the SyncroPatch 384 possible. Molecular evolution analyses were performed on resources provided by Sigma2 - the National Infrastructure for High Performance Computing and Data Storage in Norway. Figures 1–3 and Supplementary Figures 2, 3, 5–8 contain images created with biorender.com.

## Author contributions

Conceptualization, S.D.R., A.T., J.O.S.; Methodology, S.D.R., A.T., A.K., E.A.B.U., J.O.S. and I.V.; Investigation, S.D.R., J.R.D., A.T., A.K., A.M., C.I.S., V.B., A.A.W., N.B., S.J., E.A.B.U., I.V. and J.O.S.; Writing - Original Draft, S.D.R.; Writing-Review & Editing, All authors; Funding Acquisition, S.D.R., J.R.D. and J.O.S.; Resources, S.D.R., J.R.D., A.T., A.K., C.I.S., E.B., M.T., E.A.B.U, J.O.S., G.F.K. and I.V.; Supervision, S.D.R., I.V. and G.F.K.

## Competing interests

The authors declare no competing interests.
