## [Peer Review File · Nature Communications]

Ant venoms contain vertebrate-selective pain-causing sodium channel toxinsReviewers' Comments:

Reviewer #1:

Remarks to the Author:

The manuscript by Robinson et al describes a partial mechanism of action of multiple toxins isolated from ant venoms. These venoms can induce pain and the authors attribute this to toxin activity on sodium channels. The groups involved are well known for these types of projects and have a history of identifying biologically interesting toxins that have been used in sometimes game-changing publications. However, this manuscript does not belong to that category, largely due to very limited knowledge gains, strong conclusions without experimental backup, few technical issues, and messy writing.

Overall, several new peptides were identified and functional testing resulted in a link -proposed by the authors- to pain sensation. To this end, the authors mix multiple approaches including calcium imaging on F11 cells, DRG, electrophysiology on a subset of sodium channel subtypes expressed in HEK293 or CHO cells and a crude animal model of pain. Several questions arise, in no particular order of importance:

1. Why F11 cells?
2. Strong statements like claiming that a peptide is solely responsible for pain caused by a particular ant venom without taking into account other scenarios, are not appropriate.
3. Ta3a reduces calcium influx into DRG for only about 50% indicating the involvement of other factors. Yet, the authors conclude that Ta3a pain activity is dependent on sodium channels. While this may be partly true, the data does not support this statement.
4. Why were these toxins not tested on DRG sodium currents? Why were Nav1.6, 1.7, and 1.8 chosen and other sodium channel subtypes ignored? A full panel should be tested and shown.
5. Adding TTX to show that sustained sodium currents in a Nav1.7 cell line treated with toxin are due to sodium channel activity is a bizarre experiment. Claiming that current is 'via the central pore' is an amateur statement since we have known for a long time that sodium ions move through the pore of a channel. This may have been an oversight though.
6. Seemingly, random toxin concentrations were used in the reported mouse assay. What is the rationale behind these quantities? No thought was given to different pain modalities, just nocifensive responses that can be caused by multiple factors, not just peptide. Controls (such as KO mice) are completely missing.
7. Generic statements such as 'Ta3a underpins the painful stings via modulation of sodium channels in sensory neurons' are simply not true based on the data shown. First, the authors do not provide sufficient data that these toxins are sodium channel (not even subtype) specific. Second, the authors do not provide a single shred of mechanistic evidence on how these toxins work. No binding site is provided. Third, the authors execute a few simple electrophysiological protocols and ignore a whole series of others (inactivation, slow/fast, recovery...). Fourth, the authors mix species. For example, DRG were from mice, cloned channels were from humans. Without knowing the binding site, comparing these data is pure speculation. Hence, statements like the one above hold no meaning based on the data provided here.
8. The quality of several data points in relation to electrophysiology is questionable. There are signs of leaky cells, overexpression, etc...
9. The section on poneratoxin is not new. Most, if not all of the data, has been published before by other groups.
10. Pc1a/Rm4a required a double-boltzman fit. Why? Was the maximal effect not yet reached? How do these toxins work? Binding site? Other gating parameters?
11. Rm4a effects in mice were gradual in onset yet calcium response was immediate (yet only 50% inhibited by TTX). Thus, it is likely that other ion channels are targeted as well. The authors do not even acknowledge this possibility and over interpret the data here (again).
12. The authors claim that these toxins are vertebrate-specific yet provide insufficient data to support this statement. There are still effects shown.
13. The evolution section (only distracts from the rest of the paper) as well as the discussion

paragraph are poorly written. Especially the Discussion has little added value.

14. Piek's group isolated Poneratoxin (Ref 11), showed its pre- and post-synaptic interactions with different tissues/cells (Ref 12), and studied the synthetic peptide in isolated skeletal muscle (Ref 13). Later, Johnson (14) tested poneratoxin on Nav1.7. Considering that nociception is the focus of this study, toxin activity on Nav1.7 should be tested in more detail as well as a comparison to Nav1.4. .

15. Are the venom peptides the only pain-inducing substances found in (all) ant venoms? This is the impression that the reader has now. Since the authors point to membrane disruption (which I interpret as damage to membrane integrity) as a possible source of ant-sting pain, what are the other pain-inducing mechanisms besides sodium channel activation?

16. The experimental number in several assays is too low and not motivated.

17. The authors conclude that Ta3a is likely to be the primary responsible for the long-lasting pain in the *T. africanum* sting. The chromatogram of the venom shows that Ta2a may be present in higher quantities. Thus, it may be possible that upon venom injection by the ant, Ta2a still plays a substantial role in pain induction?

18. Why are single cell recordings performed only on Nav1.7 but not Nav1.6 and 1.8 (and with Ta3a)? Why are they performed at all?

19. Fig. 1m. Nav1.8 concentration-response curve is incomplete, yet an EC50 was calculated.

20. Why were these concentrations of poneratoxin, Rm4a, and Mrila chosen for in vitro assays?

Combined, this manuscript identifies new peptides that may be interesting for further studies. However, the authors fail to provide convincing evidence for a role of these peptides in causing pain. At this point, the paper reads like a combination of observations but completely lacks scientific depth. As such, this paper should be submitted to a toxin-centric journal and not a generalist journal such as *NComms*.

Reviewer #2:

Remarks to the Author:

This manuscript provides compelling evidence for a mechanism through which ant venom constituents can act on vertebrates. The primary data are diverse and significant, including the characterization of the venom, transcriptomes of the venom gland, and experimental studies of the interaction of venom constituents with predators and with cells from predators. My only concern with this work is that the authors test only those venoms that generate extreme pain, providing no negative control for their inference that the reaction is tied to the intense pain caused by the venom in these specific taxa.

The inclusion of phylogenetic perspective in interpreting the origin of the focal toxins is a strength of the work, but the methods and results are not explored with much nuance or depth. The tree they use for ants is not the most recent (Branstetter et al 2017 is probably better) and the methods for inferring the incidence of evolution of the pain-causing constituents are not explained.

Reviewer #3:

Remarks to the Author:

In this manuscript, S.D. Robinson and colleagues describe the identification and pharmacological characterization of ant toxins associated with long-lasting pain symptoms. First, the nociceptive effect of the toxins was demonstrated using pain behaviour tests. Secondly, using calcium imaging on DRG neurons and whole cell/single channel electrophysiology, the authors demonstrate that ant toxins cause pain by targeting and increasing the activity of specific Nav channel subtypes. Finally, through phylogenetic reconstruction and functional assays, the authors show that these toxins evolved early in the Formicidae as vertebrate-specific defensive agents.

This is a very interesting manuscript that address the largely unexplored question of the mode of

action of pain-causing toxins from ant venoms. The experiments are well performed and designed and the manuscript is clear and well written.

Some points need to be clarified or specified in the manuscript and additional experiments could be performed to decipher the mode of action of these toxins:

- In the process of toxins identification, the authors report some similarities between the active fractions from *T. africanum* and known toxin sequences (Table 1 and Sup Fig. 1a). It would be useful to precise how these sequences were selected (Blast analysis ?) and the % identity between them.
- Could you comment on the respective intensity and slope of the Ca²⁺ change observed in F11 cells with Ta2a and Ta3a (Fig. 1b) ?
- In the whole-cell patch clamp experiments reported in Fig. 1 (1g and 1k bold traces), can you explain why in apparent similar recording conditions, the kinetics of the current response and the effect of the toxin are so different (i.e : Ta3a increases the max current in 1g and not in 1k) ?
- It would be instructive to prepare a Table including the respective EC₅₀ values of Ta3a, Pc1a, Rm4a and Mri1a issue from voltage-clamp electrophysiology for the different Nav channels subtypes. Comparison of the different potency and selectivity profiles of the toxins with their primary structures may raise interesting questions about the molecular determinants that should be associated with their distinct pharmacological properties, such as the origin of the three-order of magnitude higher affinity of Ta3a on Nav1.7 compared to Mri1a. A preliminary SAR analysis of the results could be discussed.
- To go further, a better understanding of the mode of action of ant toxins on Nav1.7 channel as compared to previously described toxins affecting the channel inactivation (OD1 scorpion toxin interacting with site 3) should be addressed. Investigation of the effect of Ta3a on chimeric Nav-Kv channels may be useful in identifying the voltage-sensor domains involved in this interaction and in the delineation of the toxin binding site.
- Even if a correlation between in-vivo pain behaviours and in-vitro potency on Nav channels is not obvious, how to explain that Pc1a which is less potent on the three Nav channel subtypes than Ta3a induces a stronger (maximum pain behaviour counts) and faster (time to reach the plateau) pain effect (Fig. 1a and 2a) ?
- How do the authors explain that despite a very similar affinity on Nav1.6/1.7/1.8, the percentage of Pc1a and Rm4a activated DRG neurons blocked by TTX are approximately 100% and 50%, respectively ? (Sup Fig. 3c, 3f). Is it possible to test the effect of the toxins on Nav1.9 subtype ?

Minor points

- What is the rationale for using Ca imaging on both F11 and mouse DRG cells ?
- Line 101, the % of sustained peak in the presence of Ta3a is not 26% of control peak but 26% of the I_{max} with the toxin
- The concentration of Ta3a used for Nav1.7 activation is different in the text (10 nM, line 106) and in the Fig 1k-l legends (30 nM)
- How the ordinate of the Fig 1m, 2d, 2j, I_{40ms}/I_{peak}, can be greater than 1 ? The intensity of the peak at 40 ms should be smaller than the I_{max} ?
- Line 111, the authors report an EC₅₀ = 464 ± 30 nM for Nav1.8. Based on the figure 1m, it seems difficult to extrapolate a precise value ???
- Why the number of experiments for the different electrophysiological experiments are not the same. In Fig. 1m for example, n=3, 7 or 4 for Nav1.7, 1.6 and 1.8, respectively ?
- Line 187 : Supplementary Fig. 4d-e should be replaced by Supplementary Fig. 4c-d
- Line 221-223 : Results on the interaction of Mri1a toxin on Nav1.8 channel are not present in Sup. Fig. 6. Delete Sup Fig. 6f-j

- Line 273 : PD50 value should be in pmol/g. The values discussed here correspond to the administered doses (in nmol)
- In the Fig. 4, some discrepancies appeared in the toxin sequences. For example, the sequence of Poneratoxin/Pc1a in Fig.4 is not identical to that one in Fig. 1 (at the Cter part). The Tb1a sequence reported in Fig. 4 corresponds to an other toxin than the Tb1a toxin in SupFig.1 ??

REVIEWER COMMENTS

Reviewer #1 (Remarks to the Author):

The manuscript by Robinson et al describes a partial mechanism of action of multiple toxins isolated from ant venoms. These venoms can induce pain and the authors attribute this to toxin activity on sodium channels. The groups involved are well known for these types of projects and have a history of identifying biologically interesting toxins that have been used in sometimes game-changing publications. However, this manuscript does not belong to that category, largely due to very limited knowledge gains, strong conclusions without experimental backup, few technical issues, and messy writing.

We appreciate the reviewers time and effort in reviewing our manuscript. While we've made an effort to address all of the comments, some of them appear to be derived, at least in part, from misunderstandings of the rationale behind certain key experiments (see questions 1,2,4,5,8,12,13,14,15,18 below) —we have now clarified these parts of the manuscript.

In response to reviewer comments (below) we have also now included data from several additional experiments: Patch-clamp electrophysiology at mouse Nav1.7 (Supplementary Fig. 2) and human Nav1.9 (Supplementary Fig. 3) and amelioration of *in vivo* toxin-induced spontaneous pain behaviours by TTX (Supplementary Fig. 5). The results of these additional experiments are consistent with our conclusions.

Overall, several new peptides were identified and functional testing resulted in a link -proposed by the authors- to pain sensation. To this end, the authors mix multiple approaches including calcium imaging on F11 cells, DRG, electrophysiology on a subset of sodium channel subtypes expressed in HEK293 or CHO cells and a crude animal model of pain. Several questions arise, in no particular order of importance:

1. Why F11 cells?

The F11 cell line is a mouse neuroblastoma × rat dorsal root ganglion (DRG) cell line (as stated in the first section of the Results) that is often used as a model of mammalian sensory neurons. Because we hypothesised that ant stings cause pain in mammals primarily via direct local action on the peripheral sensory nervous system, these cells were chosen as a model for our high-throughput screen for pain-causing toxins of *T. africanum* venom. These results were then confirmed in cultures of primary DRG neurons and *in vivo* models of pain behaviour. Understanding the rationale behind this experiment is critical in interpreting the results, and therefore we have now clarified this in the text (lines 65-66).

2. Strong statements like claiming that a peptide is solely responsible for pain caused by a particular ant venom without taking into account other scenarios, are not appropriate.

Where these or similar conclusions are presented in the manuscript, they are supported by extensive experimental data from multiple assays. Where it could be argued that experimental support is limited, we have now reworded the sentences (lines 98-99, 138-139, 226-229, 256-257).

3. Ta3a reduces calcium influx into DRG for only about 50% indicating the involvement of other factors. Yet, the authors conclude that Ta3a pain activity is dependent on sodium channels. While this may be partly true, the data does not support this statement.

To paraphrase: [TTX inhibits Ta3a-mediated calcium influx in ~50% of DRGs neurons.] Both TTX-sensitive and TTX-resistant (Nav1.8 and Nav1.9) Nav channels are expressed in DRG neurons and Ta3a is active at both. The proportion of DRG neurons still activated in the presence of TTX is consistent with the expression of TTX-resistant Nav channels. Although we cannot claim to have tested all mammalian proteins, our data indicate no activity at other targets or transmembrane proteins expressed in neurons or HEK293 cells.

We have now included additional data that are consistent with our conclusion—i.e. amelioration of Ta3a-induced pain behaviours by injection of TTX *in vivo* (Fig. 1h). We have also now reworded the aforementioned statement as follows “...Ta3a can directly activate mammalian sensory neurons and that its mode of action in this setting involves TTX-sensitive Nav channel subtypes” (lines 98-99).

4. Why were these toxins not tested on DRG sodium currents? Why were Nav1.6, 1.7, and 1.8 chosen and other sodium channel subtypes ignored? A full panel should be tested and shown.

Our decision to test at Nav1.6, Nav1.7 and Nav1.8 is justified in the text: “Nav1.6, Nav1.7 and Nav1.8 are subtypes critical for function of mammalian DRG neurons and have each been implicated in peripheral pain signalling^{15,17}”. Nevertheless, we have now extended our analysis to include additional data for Nav1.9 (Supplemental Fig. 3 and Supplemental Table 1).

5. Adding TTX to show that sustained sodium currents in a Nav1.7 cell line treated with toxin are due to sodium channel activity is a bizarre experiment. Claiming that current is ‘via the central pore’ is an amateur statement since

we have known for a long time that sodium ions move through the pore of a channel. This may have been an oversight though.

In the Nav1.7 cell line, Ta3a caused a sustained current which does not inactivate on repolarisation. To our knowledge this effect is unique among described peptide toxins, and could reasonably be mistaken as an artefact i.e. leaky cells (as indeed may have been done by reviewer 1 here; see Q8) or pore-formation by the peptide (as is the case with some other ant venom peptides; see Introduction). Therefore the experiment demonstrating block by TTX (a toxin that blocks the pore of Nav1.7) of this leak current was necessary to conclusively demonstrate that said leak current was mediated via the central pore of the channel, rather than via toxin pore formation. To avoid further confusion, we have now clarified this sentence in the text (line 111-112).

6. Seemingly, random toxin concentrations were used in the reported mouse assay. What is the rationale behind these quantities? No thought was given to different pain modalities, just nocifensive responses that can be caused by multiple factors, not just peptide. Controls (such as KO mice) are completely missing.

The peptide concentrations used *in vivo* were based on potencies determined *in vitro* and our estimate of peptide quantities injected following envenomation (% of total venom) is consistent with the doses of synthetic peptide used in these studies.

As shown in the figures and explained in the text, negative controls (mice injected with vehicle) were used in all experiments. The nocifensive responses reported for each of the peptides clearly derive from the peptides.

7. Generic statements such as ‘Ta3a underpins the painful stings via modulation of sodium channels in sensory neurons’ are simply not true based on the data shown. First, the authors do not provide sufficient data that these toxins are sodium channel (not even subtype) specific.

The aforementioned statement has now been amended to the following: “Taken together, these data demonstrate that the peptide toxin Ta3a contributes to the long-lasting pain from stings of *T. africanum* via potent modulation of Nav channels in peripheral sensory neurons.” (lines 138-140)

Nowhere have we stated that these peptides are subtype specific, nor have we stated that they lack off-target effects at receptors other than Nav channels. However our data clearly point to Nav channel modulation playing a key role in the nocifensive effects of these toxins.

Second, the authors do not provide a single shred of mechanistic evidence on how these toxins work. No binding site is provided.

We provide extensive data from multiple assays demonstrating that these peptides directly target and cause gain-of-function at Nav channels that have well-established roles in nociception.

Demonstration of a binding site is not a prerequisite for showing that these peptides act on Nav channels (which our data clearly demonstrate that they do). The determination of a toxin’s specific binding site represents an extensive body of work in itself and will be the focus of a later manuscript.

Third, the authors execute a few simple electrophysiological protocols and ignore a whole series of others (inactivation, slow/fast, recovery...).

The electrophysiological protocols used in this study are sufficient to clearly demonstrate a gain-of-function in Nav channels in the presence of toxins. While feasible, the additional electrophysiology experiments suggested by the reviewer would add little information in this context. While we agree that the biophysical effects induced by Ta3a are intriguing, detailed electrophysiological characterisation or molecular biology experiments are beyond the scope of this work, which focused on the evolution of pain-causing toxins in ant venoms.

Fourth, the authors mix species. For example, DRG were from mice, cloned channels were from humans. Without knowing the binding site, comparing these data is pure speculation. Hence, statements like the one above hold no meaning based on the data provided here.

The overall goal of this study was to understand the mechanism of pain in humans. The assays that we were not able to perform in humans or human cells, were performed in mouse models, as is common practice. The experiments that we could do in a human system i.e. electrophysiology of human Nav channels, were performed as such. Humans and mice share homologous Nav channels (with 93% sequence similarity) and major differences in toxin effects between human and mouse channels are rare. Nevertheless, we have now performed additional electrophysiology experiments on mouse Nav1.7 (Supplementary Fig. 2), demonstrating comparable effects of these toxins at Nav1.7 of mice and humans.

8. The quality of several data points in relation to electrophysiology is questionable. There are signs of leaky cells, overexpression, etc...

Our data clearly demonstrate that the “leak” observed in Ta3a-treated Nav1.7 is an effect of the toxin rather than an artefact of suboptimal recordings (see response to Q5 above).

9. The section on poneratoxin is not new. Most, if not all of the data, has been published before by other groups. This is not correct. We are not aware of any publications reporting blockade by TTX of poneratoxin effects on DRG neurons, electrophysiological analyses of poneratoxin effects on human Nav1.6 or Nav1.8, or *in vivo* pain behaviour studies with poneratoxin. In the instances where we replicated experiments performed in previous studies (i.e., electrophysiological experiments of poneratoxin on Nav1.7; previously reported by Johnson et al (ref. 14)), which was necessary for direct comparison to the other channel subtypes, this is clearly stated.

10. Pc1a/Rm4a required a double-boltzman fit. Why? Was the maximal effect not yet reached? How do these toxins work? Binding site? Other gating parameters?

We have not presented an explanation for the observed double-Boltzmann fit of the Nav1.7 G-V curves for Pc1a and Rm4a in this manuscript. The goal of the electrophysiology experiments in this study was to test for toxin-induced gain-of-function. The Boltzman fits, whether single or double, clearly demonstrate a gain-of-function effect of the toxins.

11. Rm4a effects in mice were gradual in onset yet calcium response was immediate (yet only 50% inhibited by TTX). Thus, it is likely that other ion channels are targeted as well. The authors do not even acknowledge this possibility and over interpret the data here (again).

There are several possible explanations for these observations. One explanation is that in mouse DRG neurons, ion channels other than TTX-sensitive Nav channels are modulated by Rm4a. Indeed, we demonstrated that Rm4a can also modulate TTX-resistant Nav channels Nav1.8 (and Nav1.9). This does not conflict with our conclusion that the nociceptive effects of this toxin are driven largely by modulation of Nav channels.

Regarding the differences in the onset of effects *in vitro* versus *in vivo*: These are different assays and follow different time courses. In the *in vitro* experiments, the toxin has immediate, direct access to the neurons. This is not the case for the *in vivo* experiments where the time-course of toxin action is dependent on pharmacokinetic factors and involves multiple additional biological processes, including signal propagation, signal transmission in the spinal cord, as well as central processing/integration.

12. The authors claim that these toxins are vertebrate-specific yet provide insufficient data to support this statement. There are still effects shown.

Our data clearly demonstrate that the toxins (Ta3a, Rm4a and Pc1a) are substantially more potent in vertebrate models compared with invertebrate models (in biologically relevant assays). At all but the highest doses tested (for Ta3a and Pc1a), these peptides are remarkably vertebrate-specific, and Rm4a is vertebrate-specific even at the highest dose tested. Nevertheless, to address the reviewers concern here, we have changed all instances of the term “vertebrate-specific” to “vertebrate-selective”.

13. The evolution section (only distracts from the rest of the paper) as well as the discussion paragraph are poorly written. Especially the Discussion has little added value.

The importance of the evolution section was appreciated by the other reviewers and has been left in. The Discussion has now been modified according to the suggestions by reviewer 3.

14. Piek's group isolated Poneratoxin (Ref 11), showed its pre- and post-synaptic interactions with different tissues/cells (Ref 12), and studied the synthetic peptide in isolated skeletal muscle (Ref 13). Later, Johnson (14) tested poneratoxin on Nav1.7. Considering that nociception is the focus of this study, toxin activity on Nav1.7 should be tested in more detail as well as a comparison to Nav1.4. .

Yes, nociception is the focus of this study, and accordingly we focused on Nav1.7, Nav1.6 and Nav1.8 (and we now include new data on Nav1.9), which are relevant to sensory neuron function. Importantly, we also included multiple *in vitro* and *in vivo* nociception assays.

15. Are the venom peptides the only pain-inducing substances found in (all) ant venoms? This is the impression that the reader has now. Since the authors point to membrane disruption (which I interpret as damage to membrane integrity) as a possible source of ant-sting pain, what are the other pain-inducing mechanisms besides sodium channel activation?

We are proposing that this is the case for *certain* ant species whose stings cause long-lasting pain (see the first line of the abstract). This is made clear throughout the manuscript (e.g. see the following sentences of the Introduction: “Membrane disruption by these amphipathic peptide toxins may be responsible for the pain of some hymenopteran stings⁷, including certain ants^{4,5}. However, these observations do not satisfactorily explain the long-lasting and characteristic sting symptoms of some ant species.” etc.). The focus of this study was to identify the toxins responsible for the *long-lasting* pain of the stings of *certain* species.

16. The experimental number in several assays is too low and not motivated.

Experimental numbers for all experiments were appropriate for the statistical tests performed.

17. The authors conclude that Ta3a is likely to be the primary responsible for the long-lasting pain in the *T. africanum* sting. The chromatogram of the venom shows that Ta2a may be present in higher quantities. Thus, it may be possible that upon venom injection by the ant, Ta2a still plays a substantial role in pain induction?

Yes, on the basis of our data Ta2a is likely present in the venom of *T. africanum* at a greater concentration than Ta3a. We show that Ta2a was able to activate mammalian sensory neurons, but was 3000-fold less potent than Ta3a (Fig. 1). Furthermore, at a 100-fold greater dose than that at which Ta3a caused long-lasting spontaneous pain behaviour *in vivo*, Ta2a did not (Fig. 1). While we cannot completely rule out that Ta2a plays some role in induction of long-lasting pain, our data clearly implicate Ta3a as the primary player, and our conclusions reflect this.

18. Why are single cell recordings performed only on Nav1.7 but not Nav1.6 and 1.8 (and with Ta3a)? Why are they performed at all?

To clarify, we performed single *channel* recordings of Nav1.7, with and without Ta3a. These experiments, using Nav1.7 as the exemplar channel, provided complementary evidence for the direct action of Ta3a at Nav1.7 and defined the mechanism of action of the peptide at a single channel level.

19. Fig. 1m. Nav1.8 concentration-response curve is incomplete, yet an EC50 was calculated.

We have now amended this figure to show the complete Ta3a concentration-response curve at Nav1.8. Note that according to the suggestion by reviewer 3, we have now changed the way these curves are generated such that we can constrain the top to < 1 (i.e. I_{max}), making it reasonable to estimate an EC_{50} on the basis of these data.

20. Why were these concentrations of poneratoxin, Rm4a, and Mrila chosen for in vitro assays?

For experiments where a single toxin concentration was used (e.g. patch-clamp *I-V* experiments), we used a concentration estimated (on the basis of concentration-response data) to cause a maximum or near-maximum effect.

Combined, this manuscript identifies new peptides that may be interesting for further studies. However, the authors fail to provide convincing evidence for a role of these peptides in causing pain. At this point, the paper reads like a combination of observations but completely lacks scientific depth. As such, this paper should be submitted to a toxin-centric journal and not a generalist journal such as NComms.

Please see our responses above and the comments from reviewers 2 and 3.

Reviewer #2 (Remarks to the Author):

This manuscript provides compelling evidence for a mechanism through which ant venom constituents can act on vertebrates. The primary data are diverse and significant, including the characterization of the venom, transcriptomes of the venom gland, and experimental studies of the interaction of venom constituents with predators and with cells from predators. My only concern with this work is that the authors test only those venoms that generate extreme pain, providing no negative control for their inference that the reaction is tied to the intense pain caused by the venom in these specific taxa.

By searching venom and genomic datasets across the Formicidae (and broader Hymenoptera) we are able to convincingly demonstrate an association between the presence of these toxins and reports of long-lasting and characteristic sting pain. Our analyses also demonstrate the converse association between the absence of these toxins and the absence of these characteristic sting symptoms. We have now added the following sentence to the text to convey this point: "Thus, this class of Nav channel toxins are found in the venoms of representatives of subfamilies known to cause long-lasting and characteristic sting symptoms e.g. Paraponerinae, Myrmicinae and Ectatomminae, and are absent from the venoms of representatives of subfamilies that cause "short and sharp" sting symptoms" e.g. Myrmeciinae and Ponerinae." (lines 350-354)

The inclusion of phylogenetic perspective in interpreting the origin of the focal toxins is a strength of the work, but the methods and results are not explored with much nuance or depth. The tree they use for ants is not the most recent (Branstetter et al 2017 is probably better) and the methods for inferring the incidence of evolution of the pain-causing constituents are not explained.

As suggested, we have now replaced the phylogenetic tree in Fig. 4 (now Fig. 5) with a simplified version of that by Branstetter et al., 2017, and included a brief description in the Methods section (lines 642-643).

We have now used a most-parsimonious character state reconstruction using accelerated transformation optimization to assess the origins of Nav channel toxins in the venoms of the Formicidae. The results are in agreement

with our conclusion. We have included a description of this analysis in the Methods section (lines 644-653; and Supplementary file 1).

Reviewer #3 (Remarks to the Author):

In this manuscript, S.D. Robinson and colleagues describe the identification and pharmacological characterization of ant toxins associated with long-lasting pain symptoms. First, the nociceptive effect of the toxins was demonstrated using pain behaviour tests. Secondly, using calcium imaging on DRG neurons and whole cell/single channel electrophysiology, the authors demonstrate that ant toxins cause pain by targeting and increasing the activity of specific Nav channel subtypes. Finally, through phylogenetic reconstruction and functional assays, the authors show that these toxins evolved early in the Formicidae as vertebrate-specific defensive agents.

This is a very interesting manuscript that address the largely unexplored question of the mode of action of pain-causing toxins from ant venoms. The experiments are well performed and designed and the manuscript is clear and well written.

Some points need to be clarified or specified in the manuscript and additional experiments could be performed to decipher the mode of action of these toxins:

In the process of toxins identification, the authors report some similarities between the active fractions from *T. africanum* and known toxin sequences (Table 1 and Sup Fig. 1a). It would be useful to precise how these sequences were selected (Blast analysis ?) and the % identity between them.

We've now added % sequence identity between Ta2a and M-MYRTX-Tb1a to the text (line 76). An alignment of Ta2a with M-MYRTX-Tb1a is presented in Fig. 1a. An alignment of Ta3a with related peptides is shown in Fig. 5a.

Could you comment on the respective intensity and slope of the Ca²⁺ change observed in F11 cells with Ta2a and Ta3a (Fig. 1b)?

We've now included the raw traces for these data in Fig 1, updated the figure legend, and included a description in the text (lines 71-73).

In the whole-cell patch clamp experiments reported in Fig. 1 (1g and 1k bold traces), can you explained why in apparent similar recording conditions, the kinetics of the current response and the effect of the toxin are so different (i.e : Ta3a increases the max current in 1g and not in 1k)?

Yes — the x-axis was labelled incorrectly for panel k, it should be 1 ms, not 10 ms (as in panel g). This has now been corrected (Fig. 2 panel e and also in panels b, e, h and k of Fig. 3)—thank you for noting this discrepancy.

We have also changed the overlay in Fig. 1g so that the control trace is not masked by the Ta3a trace.

It would be instructive to prepare a Table including the respective EC₅₀ values of Ta3a, Pc1a, Rm4a and Mri1a issue from voltage-clamp electrophysiology for the different Nav channels subtypes. Comparison of the different potency and selectivity profiles of the toxins with their primary structures may raise interesting questions about the molecular determinants that should be associated with their distinct pharmacological properties, such as the origin of the three-order of magnitude higher affinity of Ta3a on Nav1.7 compared to Mri1a. A preliminary SAR analysis of the results could be discussed.

We agree that the SAR of this toxin family will be of considerable interest. The suggested table is now included as Supplementary Table 1. We are currently conducting detailed SAR studies of this toxin family and this will be the subject of a later publication. Due to space limitations we have refrained from including a detailed discussion of peptide SAR in the present manuscript.

To go further, a better understanding of the mode of action of ant toxins on Nav1.7 channel as compared to previously described toxins affecting the channel inactivation (OD1 scorpion toxin interacting with site 3) should be addressed. Investigation of the effect of Ta3a on chimeric Nav-Kv channels may be useful in identifying the voltage-sensor domains involved in this interaction and in the delineation of the toxin binding site.

We have now added a paragraph to the Discussion that briefly compares these toxins to other Nav channel toxins (lines 403-414). We agree that identification of the toxin binding site will be of interest. This is something we have begun exploring, but we hope the reviewer can appreciate that this represents a substantial body of work in its own right, and as such we plan to present this in a separate publication.

Even if a correlation between in-vivo pain behaviours and in-vitro potency on Nav channels is not obvious, how to explain that Pc1a which is less potent on the three Nav channel subtypes than Ta3a induces a stronger (maximum pain behaviour counts) and faster (time to reach the plateau) pain effect (Fig. 1a and 2a) ?

Yes, this is an interesting observation. The major difference between the effects of the two toxins *in vivo* is the *onset* of spontaneous pain behaviours i.e. the effect of Pc1a is almost immediate whereas that of Ta3a (and Rm4a) is much more gradual. There are several factors that may contribute to these differences, including pharmacokinetic-pharmacodynamic effects that may affect *in vivo* access of these peptides to cutaneous nerve terminals. However, as we have not explicitly tested this, we have refrained from any speculation in the present manuscript.

How do the authors explain that despite a very similar affinity on Nav1.6/1.7/1.8, the percentage of Pc1a and Rm4a activated DRG neurons blocked by TTX are approximately 100% and 50%, respectively? (Sup Fig. 3c, 3f). Is it possible to test the effect of the toxins on Nav1.9 subtype?

Under patch clamp conditions, hNav1.8 is more sensitive to Rm4a (and Ta3a) than Pc1a and the proportion of neurons (40-50%) still activated by Rm4a (and Ta3a) in the presence of TTX is consistent with expression of Nav1.8 in DRGs. We have now performed additional experiments with hNav1.9, demonstrating activity of the peptides at this channel subtype (Supplementary Fig. 3). We speculate that the 40-50% neurons still activated by Rm4a (and Ta3a) in the presence of TTX reflect expression of mNav1.8 (and possibly mNav1.9) and that under these conditions (i.e. calcium imaging of mouse DRG culture) there may be a greater difference in the sensitivity of these channels to Rm4a versus Pc1a (e.g. species difference, subunit expression, channel state), but cannot rule out the possible contribution of a non-Nav channel/receptor(s). Regardless, our major conclusions—that these toxins act on Nav channels and that this action is the primary mediator of the observed *in vivo* pain behaviours—remain well-supported by our data.

Note: Due to a retracted commercial agreement, we were not able to perform more experiments than those presented at Nav1.9.

Minor points

- What is the rationale for using Ca imaging on both F11 and mouse DRG cells?

Both cell types can be considered as *in vitro* models of nociception. Our primary DRG cultures are not suitable for high-throughput assays (e.g. for screening fractions using the FLIPR) and therefore we used F11 cells for this experiment. We then used the primary DRG cultures to explore the mechanism of action of the toxins.

- Line 101, the % of sustained peak in the presence of Ta3a is not 26% of control peak but 26% of the I_{max} with the toxin

It should be 26% of the control peak. We have now altered the overlay of the traces to make this clearer (Fig. 2a).

- The concentration of Ta3a used for Nav1.7 activation is different in the text (10 nM, line 106) and in the Fig 1k-l legends (30 nM)

Thank you. The figure legend has now been corrected (10 nM).

- How the ordonnate of the Fig 1m, 2d, 2j, I_{40ms}/I_{peak} , can be greater than 1? The intensity of the peak at 40 ms should be smaller than the I_{max} ?

As suggested, we have now changed these analyses to I_{40ms}/I_{max} throughout (previously $I_{40ms}/I_{peak(control)}$). This is reflected in small shifts in the reported EC_{50} values.

- Line 111, the authors report an $EC_{50} = 464 \pm 30$ nM for Nav1.8. Based on the figure 1m, it seems difficult to extrapolate a precise value ???

At higher concentrations of Ta3a, Nav1.8-expressing cells frequently lost seal and we were unable to record consistent data. We now use I_{40ms}/I_{max} for our concentration-response data. Using this approach we can constrain the top of the curve to < 1 , which results in a good fit to our data and a reasonable estimate for the EC_{50} .

- Why the number of experiments for the different electrophysiological experiments are not the same. In Fig. 1m for example, $n=3, 7$ or 4 for Nav1.7, 1.6 and 1.8, respectively?

These experiments were performed using automated patch clamp, where the number of successfully patched cells per plate varies.

- Line 187 : Supplementary Fig. 4d-e should be replaced by Supplementary Fig. 4c-d

Thank you. This has now been amended.

- Line 221-223 : Results on the interaction of Mri1a toxin on Nav1.8 channel are not present in Sup. Fig. 6. Delete Sup Fig. 6f-j

Thank you. This has now been amended.

- Line 273 : PD_{50} value should be in pmol/g. The values discussed here correspond to the administered doses (in nmol)

Because we are testing a local effect in the pain behaviour assays, we use amount (in nmol). For the insect assay, we are testing a systemic effect and we use toxin dose (in nmol/g). In order to compare the two “doses”, as is done in this paragraph, it was necessary to convert the latter to an *amount* (in nmol). We have now reworded the paragraph to make this clearer.

- In the Fig. 4, some discrepancies appeared in the toxin sequences. For example, the sequence of Poneratoxin/Pc1a in Fig.4 is not identical to that one in Fig. 1 (at the Cter part). The Tb1a sequence reported in Fig. 4 corresponds to an other toxin than the Tb1a toxin in SupFig.1 ??

“Poneratoxin” was the name given to the peptide originally isolated by Piek and colleagues. Since then a number of paralogues have been reported. We used the original poneratoxin (sequence shown in Table 1), which we’ve called Pc1a, for our studies, so that our data were comparable with the studies of Piek (Ref. 12), Duval (Ref. 13) and Johnson (Ref. 14). The prepropeptide sequence for the original Pc1a has not been reported, only that of the paralogue Pc1e, so we have used this for the alignment in Fig. 5 (previously Fig. 4).

Tb1a in Fig. 5 refers to U₃-MYRTX-Tb1a, while Tb1a in Fig. S1 refers a different peptide M-MYRTX-Tb1a. We’ve now updated the names for poneratoxin and the “Tb1a” sequences in Fig. 5.

Reviewers' Comments:

Reviewer #3:

Remarks to the Author:

All of my concerns have been adequately addressed in the revised manuscript and the authors have included several additional experiments strengthening the consistency of their conclusions. In addition, the changes introduced in the text have clarified the manuscript which can now be accepted for publication.

Reviewer #4:

Remarks to the Author:

The manuscript entitled "Ant venoms contain vertebrate-selective pain-causing sodium channel toxins" describes the putative effects of sodium channel modulation by select peptides found in ant venoms. The idea here is that ants developed these peptides as a defense mechanism to inflict pain on predators. The authors use a myriad of techniques including peptide synthesis, patch clamp, calcium imaging and pain behavior. Overall, I tend to agree with reviewer 1 that this manuscript would be more appropriate for a toxin or a pain specific journal. The results are more observational, and at times superficial, while lacking strong mechanistic insight. Moreover, the data at times conflicts with each other leading to more confusion than clarity.

1. I am mostly perplexed by Figs 2a, 2e, 2h and 2j as they relate to each other. First, in Fig 2a the peak current is larger with Tac3a compared to control, but fast inactivation is completely identical. Afterwards fast inactivation, the appearance of the sustained current appears. Meanwhile for hNav1.7 in Fig 2E, the peak current is smaller with Tac3a, and again though the channels are still inactivating. Why the discrepancy in peak current between Fig 2a and 2e? (also pointed out by Reviewer 3) Reviewer 1 called this a "leaky" cell, and perhaps the better way of addressing it, is to say that the integrity of the patch is compromised. This would result in difficulties in maintaining the voltage clamp and could explain both the decrease in peak current and the observance of a leak current. As we get to the single channel data, this is where it's unclear if Tac3a is indeed affecting Nav1.7 channels or alternatively affecting some endogenous channel that exists in this stable cell line. Because at -20 mV, Nav1.7 should be completely inactivated (PMID: 26068619). It's even surprising that they observe open events of 0.1 at this voltage. Nonetheless, this data suggests Tac3a completely uncouples inactivation, but we should see this early in the macroscopic currents in Figs 2a and e, but we do not. Reviewer 1 had asked for more complete electrophysiological analyses (inactivation, slow/fast, recover) but the authors have opted not to address this but it's important to do so. Again, the other interpretation is that Tac3a might be affecting some endogenous channel. The experiment to confirm Nav1.7 is to perfuse TTX directly onto patches. That was not done here. The experiment incubating Tac3a on untransfected HEK cells is not a definitive control experiment. Because untransfected HEK cells aren't comparable to this stably Nav1.7 transfected cell line.
2. There are no wash experiments. This is an absolute must with such low n values. These are soluble peptides, and the effects should be at least partially reversible if not fully reversible after wash. This would help reaffirm the integrity of the electrophysiological data.
3. Calcium imaging measures many different things. The most convincing data on the electrogenic/sodium channel properties of these peptides is to assess DRG neuronal excitability directly. I think this is also a must experiment. If the authors claim that peptides are affecting sodium channels, then this is best assessed by current clamping DRG neurons and examining various measures of excitability (Vm, rheobase, action potential height and width, loss of firing accommodation).
4. It's strange that nocifensive responses are not measured more than 30 minutes. Going beyond 30 minutes helps understand the duration of the pain response (return to baseline as the peptides are cleaved up by proteases). Strikingly there are no statistical analyses provided in Figs 1d, 3a and 3g. These data would have to be assessed by ANOVA.

5. A scrambled peptide is a more appropriate control than vehicle.
6. Is the calcium imaging data in Fig 1 e in DRG neurons (n=3) isolated from one mouse? If so, that's a technical replicate and not a biological replicate.
7. What are the n values for Fig 1d?

RESPONSE TO REVIEWER COMMENTS

Reviewer #3 (Remarks to the Author):

All of my concerns have been adequately addressed in the revised manuscript and the authors have included several additional experiments strengthening the consistency of their conclusions. In addition, the changes introduced in the text have clarified the manuscript which can now be accepted for publication.

Reviewer #4 (Remarks to the Author):

The manuscript entitled “Ant venoms contain vertebrate-selective pain-causing sodium channel toxins” describes the putative effects of sodium channel modulation by select peptides found in ant venoms. The idea here is that ants developed these peptides as a defense mechanism to inflict pain on predators. The authors use a myriad of techniques including peptide synthesis, patch clamp, calcium imaging and pain behavior. Overall, I tend to agree with reviewer 1 that this manuscript would be more appropriate for a toxin or a pain specific journal. The results are more observational, and at times superficial, while lacking strong mechanistic insight. Moreover, the data at times conflicts with each other leading to more confusion than clarity.

Please see comments below.

1. I am mostly perplexed by Figs 2a, 2e, 2h and 2j as they relate to each other. First, in Fig 2a the peak current is larger with Tac3a compared to control, but fast inactivation is completely identical. Afterwards fast inactivation, the appearance of the sustained current appears. Meanwhile for hNav1.7 in Fig 2E, the peak current is smaller with Tac3a, and again though the channels are still inactivating. Why the discrepancy in peak current between Fig 2a and 2e? (also pointed out by Reviewer 3)

The data shown in panels 2a and 2e are derived from different experimental protocols (the pulse protocols are shown in each panel and described in the figure legend and corresponding Methods section) and there should not be an expectation that they would be identical. The increase in non-inactivating current caused by Ta3a, which is matched by a decrease in peak current, is time-dependent. In the I-V protocol (Fig 2e) the cell is unavoidably exposed to Ta3a for a longer time than in the single pulse protocol (Fig 2a), hence the difference in traces. Figs 2h and j relate to the single channel recordings (see response below).

Reviewer 1 called this a “leaky” cell, and perhaps the better way of addressing it, is to say that the integrity of the patch is compromised. This would result in difficulties in maintaining the voltage clamp and could explain both the decrease in peak current and the observance of a leak current.

We realize that the effects of Ta3a are distinct from almost all previously described Nav modulators and that on the surface they could be interpreted as being derived from “leaky” or “compromised” cells. However our experiments clearly demonstrate that Ta3a causes a non-inactivating current that is mediated via the central pore of Nav1.7: Ta3a does not induce the non-inactivating current in non-transfected HEK293 cells (Fig 2c) and the non-inactivating current in Nav1.7 cells can be blocked by TTX (Fig 2b,d). In addition the effects are concentration-dependent, voltage-dependent and highly reproducible i.e. consistent across multiple cells and multiple experiments. Furthermore, Figure 3 of this same manuscript (as well as numerous other publications) illustrates our capacity to acquire and correctly interpret high quality electrophysiology data for more conventional peptide toxins.

As we get to the single channel data, this is where it’s unclear if Tac3a is indeed affecting Nav1.7 channels or alternatively affecting some endogenous channel that exists in this stable cell line. Because at -20 mV, Nav1.7 should be completely inactivated (PMID: 26068619). It’s even surprising that they observe open events of 0.1 at this voltage. Nonetheless, this data suggests Tac3a completely uncouples inactivation, but we should see this early in the macroscopic currents in Figs 2a and e, but we do not.

The single channel voltage protocol from a holding potential of -110 mV to -20 mV was chosen because this voltage produces maximal Nav1.7 activity, as shown in the whole-cell I-Vs presented in the Figure 2f. Other papers that perform a similar voltage step protocol produce comparable single channel activity to our data presented in Figure 2h (PMID: 26646206). In addition, the intracellular solution that we used in the single channel experiments contains 150 mM Cs⁺ ions (see Methods section), which will block any low level expression of K⁺ channels in our membrane patches.

The single channel activity in the presence of Ta3a is of a single ion channel that is affected by the peptide. By contrast, the whole-cell responses indicate that at an early stage of the experiment not all available channels are affected by Ta3a and therefore retain normal activation properties, while other channels that are affected by Ta3a become persistently active. This produces the two components of the whole-cell (ensemble) I-V – the peak current and the persistent current.

Reviewer 1 had asked for more complete electrophysiological analyses (inactivation, slow/fast, recover) but the authors have opted not to address this but it's important to do so. Again, the other interpretation is that Tac3a might be affecting some endogenous channel.

The effects of Ta3a on Nav currents are voltage-dependent, blocked by TTX and not present in non-transfected HEK cells (i.e. in the absence of Nav channels). These data are not consistent with the possibility of the effects being mediated by a hypothetical “endogenous channel” in our experiments. The additional electrophysiological experiments requested by the reviewer do not change the key findings/conclusion of this study. Nevertheless, we have now included steady-state fast inactivation experiments and washout experiments (figs 2 and 3).

The experiment to confirm Nav1.7 is to perfuse TTX directly onto patches. That was not done here. *This experiment was done (see Fig 2b and d), and clearly shows that the effects of Ta3a on Nav1.7 current are blocked by TTX.*

The experiment incubating Tac3a on untransfected HEK cells is not a definitive control experiment. Because untransfected HEK cells aren't comparable to this stably Nav1.7 transfected cell line.

Here we tested whether Ta3a produced a non-inactivating current in the absence of Nav channels (in non-transfected HEK cells)—it did not. This experiment demonstrated that Ta3a is not a pore-forming toxin and that it does not cause a non-inactivating in the absence of a Nav channel.

2. There are no wash experiments. This is an absolute must with such low n values. These are soluble peptides, and the effects should be at least partially reversible if not fully reversible after wash. This would help reaffirm the integrity of the electrophysiological data.

The peptide toxins presented in this study are in fact very hydrophobic (see Table 1) and poorly soluble in aqueous solution (we have now added a note on this in the Methods section). Note: Many venom peptides (including these) cause effects that are weakly or not reversible even after extensive washing (see e.g. PMID: 32938666, PMID: 21652775). Nevertheless, we have performed the suggested washout experiments for the peptides (Ta3a, Rm4a, Pc1a) at Nav channels and have now included these data in the manuscript. All experiments performed in this study were conducted with appropriate n for the statistical tests used.

3. Calcium imaging measures many different things. The most convincing data on the electrogenic/sodium channel properties of these peptides is to assess DRG neuronal excitability directly. I think this is also a must experiment. If the authors claim that peptides are affecting sodium channels, then this is best assessed by current clamping DRG neurons and examining various measures of excitability (Vm, rheobase, action potential height and width, loss of firing accommodation).

In this study we used calcium imaging experiments to narrow down the receptor(s) for these toxins in mouse DRG neurons. Our data indicated that the toxins cause an increase in intracellular calcium in DRG neurons which was blocked by TTX, thus implicating TTX-sensitive Nav channels as the probable receptors. We subsequently confirmed activity at Nav channels using voltage-clamp electrophysiology. Our data unequivocally demonstrated that these peptides affect Nav channels.

Note: Our data do not preclude activity of these peptides at other ion channels/receptors, but do demonstrate that these are the receptor(s) for these peptides in the context of pain (and our conclusions are made accordingly). Profiling the selectivity of these peptides across diverse ion channels/receptors is beyond the scope of the present study.

4. It's strange that nocifensive responses are not measured more than 30 minutes. Going beyond 30 minutes helps understand the duration of the pain response (return to baseline as the peptides are cleaved up by proteases).

We used spontaneous pain behaviour experiments to assess whether the peptide toxins caused acute pain behaviours in mice following intraplantar injection—the experiments were designed to mimic as closely as possible a stinging event. We did not design our experiments to test the longer-term duration of the pain responses beyond 30-min and our conclusions have been made accordingly.

Strikingly there are no statistical analyses provided in Figs 1d, 3a and 3g. These data would have to be assessed by ANOVA.

Analysis of these data by ANOVA is now included. The conclusions remain unchanged.

5. A scrambled peptide is a more appropriate control than vehicle.

The use of vehicle (saline + 0.1% BSA) as a negative control was appropriate for these experiments.

6. Is the calcium imaging data in Fig 1 e in DRG neurons (n=3) isolated from one mouse? If so, that's a technical replicate and not a biological replicate.

Calcium imaging data were derived from multiple mice i.e. biological replicates.

7. What are the n values for Fig 1d?

n values for figure 1d are now added to the text and figure legend.

Reviewers' Comments:

Reviewer #4:

Remarks to the Author:

I appreciate the modifications made by the authors to the manuscript, including the added wash experiments and statistical analyses for behavior. The data in Fig 2 still does not convince me that Ta3a directly affects NaV1.7 channel inactivation. In their rebuttal, the authors cite PMID 26646206, where scorpion toxins, in a concentration-dependent manner, affect NaV1.7 inactivation. In PMID 26646206, there is a slowing of fast inactivation and no effect on peak current. The data presented in this manuscript doesn't look like that. Furthermore, the authors refuse to conduct excitability measurements in DRG neurons.

1. With respect to the Fig.2a vs 2e, the explanation provided by the authors is inadequate. First, the figure legend is mislabeled. Fig 2a is mNaV1.7 not hNaV1.7. In Fig 2a, the concentration of Ta3a is 3-fold higher than in Fig2e. Thus, this time dependent effect on peak current doesn't hold water. Are the peptides are simply pipetted in by bolus? If they are not perfused one cannot assess time dependence especially at different concentrations. If there is a time dependence effect on peak current, the authors should perform the experiment. Otherwise, the reader will undoubtedly be confused by these figures. Moreover, prior criticism arose from the various concentrations of Ta3a used with no explanation as to why they are varied. This is just sloppy for a paper

2. The single channel traces shown in this manuscript and the ones referenced in PMID 26646206 are not at all similar. In PMID 26646206, sample durations are in the tens of milliseconds, while those depicted in the manuscript are in the many hundreds of milliseconds. There are channel openings in Fig 2i at -20 mV hundreds of milliseconds after the depolarization. That doesn't make sense. They should all be inactivated. Indeed, referencing PMID 26646206 makes the data in Fig 2i less convincing that these are NaV1.7 and not some other channel.

3. TTX perfusion onto excised outside-out patches **was not** done. This would prove that in single channel patches those channels are indeed NaV1.7 channels.

3. For a persistent current that is mediated through NaV1.7 to occur, it would have to uncouple inactivation. This should be seen in the macroscopic currents. The authors imply through the comment "early stage" that it is a time dependent effect. Evidence is not provided for the time dependent effect on channel inactivation.

4. Incorporation of the plasmid DNA into randomly into the HEK cell genome to produce the stable cell line might have disrupted other genes. Further, the cells are given a xenobiotic to select for the transgene. HEK cells are known to contain endogenous chloride, potassium, sodium, and calcium channels (PMID: 35567642), many are non-selective. This NaV1.7 stable cell line might have resulted in an increased expression of an endogenous channel sensitive to Ta3a. To me the data in Fig 2 suggest an endogenous channel is involved.

5. In the rebuttal, the authors claim they voltage clamped DRG neurons. I do not see where DRG neurons were voltage clamped in the manuscript. If neurons can be voltage clamped, then current clamping is completely feasible. Measuring DRG neuronal excitability in response to the peptides should be part of this manuscript, especially for consideration for publication in this journal. It is essential to reconcile issues with heterologous cells.

RESPONSE TO REVIEWER COMMENTS:

Reviewer #4 (Remarks to the Author):

I appreciate the modifications made by the authors to the manuscript, including the added wash experiments and statistical analyses for behavior. The data in Fig 2 still does not convince me that Ta3a directly affects Nav1.7 channel inactivation. In their rebuttal, the authors cite PMID 26646206, where scorpion toxins, in a concentration-dependent manner, affect Nav1.7 inactivation. In PMID 26646206, there is a slowing of fast inactivation and no effect on peak current. The data presented in this manuscript doesn't look like that. Furthermore, the authors refuse to conduct excitability measurements in DRG neurons.

We acknowledge that the effects of Ta3a on Nav1.7 current are distinct from other peptide toxins. In this way we can understand why this reviewer has focused on this aspect of the manuscript.

Nevertheless, we believe that our data clearly demonstrate that Ta3a directly affects Nav1.7 channel inactivation, and the proposed alternative hypothesis, that the observed current is due to a different unknown endogenous channel, is not supported by our data:

- *The Nav1.7 cells used is a commercial line (SB Drug Discovery) that has been validated in multiple publications. There is no published evidence of the overexpression of other channels in this cell line.*
- *Under control conditions, the observed whole cell current (amplitude, kinetics, voltage-dependence of activation etc.) in this cell line is exactly as expected of Nav1.7. There is no evidence of any current derived from any other channel.*
- *Whole cell current in this cell line is completely blocked by established highly selective blockers of Nav1.7 e.g. Pn3a (PMID: 28548111) as well as TTX.*
- *Under control conditions, the observed single channel conductance is exactly as expected of Nav1.7 (response to step depolarisation to -20 mV, unitary conductance of ~ 1.4 - 1.5 pA). There is no evidence of any current derived from any other channel.*

In short, there should not be any question that the current we are observing in this cell line, at least under control conditions, under whole cell or single channel, are derived from Nav1.7. The next question is whether the current we observe in the presence of Ta3a is also derived from Nav1.7. Again we believe that our data clearly demonstrate that this is the case:

- *The non-inactivating persistent current observed in the presence of Ta3a, is completely blocked by TTX (Fig 2b,d). This data unequivocally shows that this effect is mediated by a TTX-sensitive Nav isoform i.e. Nav1.7.*

While this result alone clearly reveals that the current observed in the presence of Ta3a is due to Nav1.7 and rules out the involvement of a hypothetical endogenous channel, there is additional evidence, as follows:

- *Cs⁺ present in the intracellular solutions for all of our electrophysiology experiments rules out the involvement of potassium channels (this is now stated explicitly in the manuscript).*
- *The effects of Ta3a are concentration-dependent and voltage-dependent and are highly reproducible i.e. consistent across multiple cells and multiple experiments.*
- *We observe identical or similar effects in multiple independently generated cell lines expressing mouse Nav1.7, hNav1.6, hNav1.9, hNav1.8—the latter of which is in a different cell (CHO) background. While it is possible that in the course of generating these cell lines, an endogenous channel(s) may have increased in expression, it is highly improbable that this would have occurred for the exact same hypothetical endogenous channel in all of these independently generated cell lines.*
- *Ta3a does not induce the persistent current in non-transfected HEK293 cells (Fig 2c), but does in cells transiently-transfected with human Nav1.7 (not included in the manuscript but shown here (10 nM Ta3a):*

In summary, our data clearly demonstrate that Ta3a directly affects $Na_v1.7$ channel inactivation and this is presented clearly in the manuscript. The alternative hypothesis that the observed current is due to a different unknown endogenous channel is not supported by these data. Because of this, we do not agree that the suggested additional experiments are necessary to support our conclusion.

1. With respect to the Fig.2a vs 2e, the explanation provided by the authors is inadequate. First, the figure legend is mislabeled. Fig 2a is mNaV1.7 not hNaV1.7. In Fig 2a, the concentration of Ta3a is 3-fold higher than in Fig2e. Thus, this time dependent effect on peak current doesn't hold water. Are the peptides are simply pipetted in by bolus? If they are not perfused one cannot assess time dependence especially at different concentrations. If there is a time dependence effect on peak current, the authors should perform the experiment. Otherwise, the reader will undoubtedly be confused by these figures. Moreover, prior criticism arose from the various concentrations of Ta3a used with no explanation as to why they are varied. This is just sloppy for a paper

Fig. 2a does not appear to be mislabelled—this is hNa_v1.7. The equivalent trace for mNa_v1.7 is shown in supplementary figure 2a.

Figure 2g shows that over the course of ~5 min of exposure to 3 nM Ta3a, the persistent current increases to a steady state where it is maintained for the remainder of the experiment (~25 min) with very little washout in-spite of repeated applications of peptide-free buffer. We used 3 nM Ta3a for this measurement because higher Ta3a concentrations reduce the lifetime of the experiment, most likely due to accumulation of non-inactivating channels and resultant changes to the properties of the cell membrane. In none of our experiments do we apply Ta3a by pipetting as a bolus into a recording chamber that is typically used in conventional patch clamp experiments. The QPatch instrument, which we used for most of the electrophysiology experiments, replaces 5 μ L of buffer solution with 5 μ L of peptide-containing buffer within 10s of milliseconds. This solution exchange time is very rapid and comparable to solution exchange times using conventional perfusion systems in standard patch clamp experiments. The slower onset of the effect of the peptide on Na_v-mediated persistent current is, therefore, not due to solution exchange times, but rather the slow mechanism of action of the peptide, which likely involves membrane partitioning, followed by peptide receptor site interactions. This relatively slow onset of persistent current is observed with 3 nM (Figure 2g), 10 nM (Figure 2j) and 30 nM (Figure 2d and e) Ta3a concentrations, and are consistent with the calcium imaging data (Figure 1f). However, we were not concerned with the kinetics or time-dependence as a function of peptide concentration in the present study, we only wished to show that the onset of persistent current is slow (Figure 2g and j) and that it is ablated by TTX (Figure 2b and d).

To avoid confusion to readers we have replaced panel a of Fig. 2 with a representative trace that is more consistent with that shown in Fig. 2e i.e. does not suggest an increase in peak current amplitude.

Note regarding peptide concentrations: The non-inactivating persistent current induced by Ta3a prevents recording over long periods. For experiments where longer recordings were required (e.g.

the IV protocol and the washout experiment), it was necessary to use submaximal concentrations of the peptide.

2. The single channel traces shown in this manuscript and the ones referenced in PMID 26646206 are not at all similar. In PMID 26646206, sample durations are in the tens of milliseconds, while those depicted in the manuscript are in the many hundreds of milliseconds. There are channel openings in Fig 2i at -20 mV hundreds of milliseconds after the depolarization. That doesn't make sense. They should all be inactivated. Indeed, referencing PMID 26646206 makes the data in Fig 2i less convincing that these are Nav1.7 and not some other channel.

As the reviewer points out, the single channel activity in PMID 26646206 is of shorter duration than our recordings by an order of magnitude. We recorded for hundreds of milliseconds to demonstrate that most of the activity occurs at the onset of the voltage step but the channels can exit inactivated states to reactivate for brief periods, particularly at -20 mV. We have shown previously that Nav1.7 and Nav1.8 isoforms do indeed reactivate if the recording is extended for longer periods using a similar activating protocol (PMID: 28225079). The activity shown in PMID 26646206 was elicited by a very similar voltage step protocol that we used in our study, has a comparable unitary current and activates mostly near the onset of the voltage step, just like our recordings – albeit it shows a much shorter recording. To better illustrate that if only the first few 10s of milliseconds of recording is shown then our recordings look the same as those in PMID 26646206 we have prepared a figure with a shorter time scale:

We generally prefer to show longer segments of single channel activity because it provides more information regarding the kinetic behaviour of Nav channels, which like all other known channels, can be modelled on a thermodynamic cycle that includes interconnected functional states. Entering a given state is probabilistic—although the likelihood of Nav channels reopening decays exponentially with a time constant of several milliseconds, all channels, (even without an activating stimulus) will activate as predicted by a Boltzmann distribution. Note that in PMID 26646206 and 28225079 (and many other studies), TTX was not applied to the single channel currents to confirm that the currents were mediated by Navs, but the data collectively in those studies provide convincing evidence that the currents were not mediated by a distinct endogenous channel.

3. TTX perfusion onto excised outside-out patches **was not** done. This would prove that in single channel patches those channels are indeed Nav1.7 channels.

We disagree that washing TTX over patches that are excised from cells that stably express Nav1.7 channels is necessary to demonstrate that the recorded channel activity is that of Nav1.7 channels. Further, we feel that it is unreasonable to suppose that 100% of patches excised from stably-expressing Nav1.7 cells will express some other endogenous channel with activation properties that are characteristic of Nav channels.

The cell line that was used for both whole-cell and single channel measurements is a validated cell line that overexpresses Nav1.7 channels with no evidence of overexpression of any other channel that might contaminate our recordings. We intentionally used a CsCl based intracellular solution to block any potential low level expression of endogenous potassium channels that have been previously reported in some HEK293 cell lines. Given that the majority of the electrophysiology data were recorded in the whole-cell configuration we felt that this was the appropriate type of recording to show TTX block. It is important to note that TTX blocked the entire whole-cell current without leaving any residual, TTX-insensitive current that might indicate a low level of expression of a hypothetical endogenous channel that cannot be blocked by Cs⁺ ions – both with and without Ta3a. We rule out the presence of voltage-activated potassium and calcium channels, which have similar current

amplitudes to Navs, but will be blocked by Cs⁺ ions (potassium channels) or not respond to TTX (potassium and calcium channels). We also rule out any endogenous channel that is not voltage-activated, because we observe no channel openings prior to depolarising steps, or with a unitary current that is not characteristic of Navs. The use of the same stably expressing cell line for all electrophysiology experiments, the response to the same activating stimulus, use of CsCl solutions and complete ablation of the whole-cell current by TTX are collectively diagnostic properties of Nav channel activity. We therefore feel that performing time-consuming single channel experiments with TTX and TTX plus Ta3a, only to re-confirm that we are recording from Nav channels would be a redundant exercise and will not add new information to the manuscript.

4. For a persistent current that is mediated through Nav1.7 to occur, it would have to uncouple inactivation. This should be seen in the macroscopic currents. The authors imply through the comment “early stage” that it is a time dependent effect. Evidence is not provided for the time dependent effect on channel inactivation.

Throughout the manuscript we have shown data that clearly demonstrates that the peptides inhibit inactivation in macroscopic currents (Figure 2a, d, e, and g, Figure 3b, f, j and n). However, the effects are not immediate and generally develop over several minutes of recording for Ta3a (Figure 2g) and Rm4a (Figure 3p), whereas the onset time for Pc1a is faster (Figure 3d). The relatively slow effect onset for Ta3a and Rm4a is not uncommon in peptides that are lipophilic and do not wash off significantly over the course of a typical recording (1–15 min), particularly Ta3a. The time-dependence of the effect is likely due to the combination of cell membrane partitioning and receptor binding site association rates of the peptides we used in the study.

5. Incorporation of the plasmid DNA into randomly into the HEK cell genome to produce the stable cell line might have disrupted other genes. Further, the cells are given a xenobiotic to select for the transgene. HEK cells are known to contain endogenous chloride, potassium, sodium, and calcium channels (PMID: 35567642), many are non-selective. This Nav1.7 stable cell line might have resulted in an increased expression of an endogenous channel sensitive to Ta3a. To me the data in Fig 2 suggest an endogenous channel is involved.

Please see our detailed response above. We see no evidence that the generation of the stable cell line has disrupted genes in the HEK cells that would have any impact on our study, and reiterate that TTX completely ablated the whole cell currents without leaving any residual current that might indicate the involvement of endogenous channels. Our data clearly demonstrate that Ta3a directly affects Nav1.7 channel inactivation. We do not claim that Ta3a is selective for Nav1.7 – indeed we already show data that Nav1.6 is also affected. However, the alternative hypothesis that the observed current is due to a different unknown endogenous channel (that is not a Nav channel) is not supported.

6. In the rebuttal, the authors claim they voltage clamped DRG neurons. I do not see where DRG neurons were voltage clamped in the manuscript. If neurons can be voltage clamped, then current clamping is completely feasible. Measuring DRG neuronal excitability in response to the peptides should be part of this manuscript, especially for consideration for publication in this journal. It is essential to reconcile issues with heterologous cells.

In the rebuttal we stated that we performed calcium imaging of DRG neurons, we did not state that we performed voltage clamp experiments of DRG neurons.

Yes, voltage clamp and current clamp recordings from DRG neurons are feasible, however we do not agree that these experiments will be useful in reconciling perceived issues with heterologous cells or indeed that there such issues exist (see detailed responses above).

Nav1.7 is indisputably the main TTX-sensitive Nav isoform expressed in DRG neurons. Inhibition of inactivation of Nav1.7 will invariably result in enhanced excitability of these neurons – this is also convincingly demonstrated by our in vivo experiments showing emergence of nocifensive responses which clearly point towards toxin-induced action potential firing.

We would like reiterate that we make no claim at any point that Ta3a is selective for Nav1.7 – it is indeed possible that the peptide has other pharmacological effects. However, as elaborated above, these peptides have clear, potent, effects on mammalian Nav channels, the mechanisms of which will not be further illuminated by the suggested current clamp experiments. This, coupled with the fact that these experiments are cost-, labour- and time-intensive, was our reason for choosing not to perform them.